# Locomotion-induced ocular motor behavior in larval *Xenopus* is developmentally tuned by visuo-vestibular reflexes

Julien Bacqué-Cazenave[1,2,3], Gilles Courtand[1], Mathieu Beraneck [4], Hans Straka[5], Denis Combes[1] & François M. Lambert[1✉]

Locomotion in vertebrates is accompanied by retinal image-stabilizing eye movements that derive from sensory-motor transformations and predictive locomotor efference copies. During development, concurrent maturation of locomotor and ocular motor proficiency depends on the structural and neuronal capacity of the motion detection systems, the propulsive elements and the computational capability for signal integration. In developing *Xenopus* larvae, we demonstrate an interactive plasticity of predictive locomotor efference copies and multi-sensory motion signals to constantly elicit dynamically adequate eye movements during swimming. During ontogeny, the neuronal integration of vestibulo- and spino-ocular reflex components progressively alters as locomotion parameters change. In young larvae, spino-ocular motor coupling attenuates concurrent angular vestibulo-ocular reflexes, while older larvae express eye movements that derive from a combination of the two components. This integrative switch depends on the locomotor pattern generator frequency, represents a stage-independent gating mechanism, and appears during ontogeny when the swim frequency naturally declines with larval age.

[1] Université de Bordeaux, CNRS UMR 5287, Institut de Neurosciences Cognitives et Intégratives d'Aquitaine, F-33076 Bordeaux, France. [2] Normandie Univ, Unicaen, CNRS, EthoS, 14000 Caen, France. [3] Univ Rennes, CNRS, EthoS (Éthologie animale et humaine)—UMR 6552, F-35000 Rennes, France. [4] Université de Paris, CNRS UMR 8002, Integrative Neuroscience and Cognition Center, F-75006 Paris, France. [5] Faculty of Biology, Ludwig-Maximilians-University Munich, Grosshadernerstr. 2, 82152 Planegg, Germany. ✉email: francois.lambert@u-bordeaux.fr

Locomotion requires concurrent neural computations that optimally maintain sensory perception despite the disturbing consequences of the motor behavior on the encoding and transformation process. Accordingly, all vertebrates optimize their visual acuity by integrating sensory feedback signals of passively-induced head/body motion with intrinsic feedforward signals during self-generated movements[1–3]. Motion-induced retinal image displacements are minimized by counteractive eye and/or head-adjustments that derive from the synergistic activity of visual, vestibular and proprioceptive reflexes[3–5], supplemented during self-motion by efference copies of the propulsive motor commands[6,7]. Independent of the locomotor style, such feedforward replicas from spinal pattern-generating circuits evoke spatio-temporally adequate eye movements as originally demonstrated in pre- and post-metamorphic amphibians[8–10]. Suggestive evidence for a ubiquitous role of such signals in all vertebrates was subsequently provided by studies in mice[11,12] and human subjects[13,14]. Gaze-stabilizing eye movements during locomotion, therefore result from the integration of locomotor efference copies and motion-related sensory signals into ocular motor commands. The computational algorithms for integrating the different signal components are likely established prior to and during the developmental acquisition of locomotion and are further modified as propulsive proficiency increases.

Progressive developmental improvements of locomotor performance are tightly linked to the maturation of gaze- and posture-stabilizing reflexes[15–17]. The structural assembly and functional initiation of inner ear sensation and associated sensory-motor computations are decisive factors for the onset and efficiency of gaze-stabilization as well as locomotion[18]. This is particularly critical in small aquatic vertebrates, where coordinated locomotor activity is generally acquired very early after hatching[15,18]. In accordance with this requirement, utriculo-ocular reflexes become functional immediately after hatching in zebrafish[19] and anuran larvae[20] followed, though conspicuously later, by semicircular canal-derived vestibulo-ocular reflexes (VOR)[16,21]. The different developmental acquisition of individual VOR components, as demonstrated for *Xenopus* larvae, but likely also of visuo-ocular reflexes, might have significant interactive influences on the maturation of locomotor proficiency with respective reciprocal consequences for efference copy-driven gaze-stabilization[8]. Since swim-related head motion in e.g., *Xenopus* tadpoles represents a reliable sensory reference, vestibular signals might be well-suited to guide the ontogenetic progression of locomotion-induced ocular motor behaviors.

Here, we demonstrate the mutual developmental plasticity of spinal locomotor efference copies and multi-sensory motion signals in the production of dynamically adequate gaze-stabilizing eye movements during swimming in larval *Xenopus*. Following simultaneous onsets of locomotor activity, spino-ocular, optokinetic and utriculo-ocular motor behaviors, efference copy-evoked predictive eye movements are gradually modified through the influence of semicircular canal signals during larval development. During this period, the sensory-motor integration changes from a restricted effect of the angular VOR on locomotor-induced compensatory eye movements in young larvae to a more robust and direct interaction between semicircular canal sensory and spino-ocular signals to elicit appropriate swim-related ocular adjustments in older larvae. The switch in computational outcome occurs in synchrony with a reduced efficacy of the tail-undulatory locomotor pattern generator during development causing a gradually decaying influence on the ocular motor output as metamorphic climax approaches.

## Results
### Common onset of predictive spino-ocular, visual motion and otolith organ-evoked eye movements. Free swimming in

*Xenopus laevis* larvae commences around stage 37/38 followed by a rapid acquisition of the typical kinematic profile that remains essentially unaltered between stage 40[22] and stage 50[9]. The early developmental occurrence of undulatory swimming, however, is at variance with the general lack of ocular motor behaviors. In fact, animals at stage 40 lack a locomotor efference copy-driven gaze-stabilizing spino-ocular motor behavior[8] (Fig. 1a₁; Supplementary Fig. 1b, $N = 3$), a horizontal optokinetic (OKR; Fig. 1b₁; Supplementary Fig. 1b, $N = 5$) as well as a gravitoinertial vestibulo-ocular reflex (gVOR; Fig. 1c₁; Supplementary Fig. 1b, $N = 5$). The earliest activation of reliable and robust compensatory eye movements, triggered by ascending copies of spinal locomotor signals, was only observed at stage 42 and had a gain of $0.34 \pm 0.08$ ($N = 3$; Fig. 1a₂; Supplementary Fig. 1b). This onset coincided with the developmental acquisition of sensory stimulus-driven visuo- and gravitoinertial (otolith organ) ocular motor behaviors. First evidence for their presence was encountered at stage 42 (Fig. 1b₂, c₂), where the OKR and gVOR consisted of small, yet reliable eye movements with gains that exceeded 0.15 ($0.16 \pm 0.02$ for OKR, $N = 5$ and $0.18 \pm 0.05$ for gVOR, $N = 7$; Supplementary Fig. 1b). Thereafter, these gaze-stabilizing eye movement components increased further in robustness to reach gains at stage 45 ($N = 4$) of $0.65 \pm 0.02$ for the spino-ocular motor coupling, $0.33 \pm 0.08$ for the OKR and $0.47 \pm 0.21$ for the gVOR (Fig. 1a₃–c₃; Supplementary Fig. 1b). The performance continued to improve until maximal gain values were reached at stage 48 with a generally better performance of the gVOR compared to the OKR (Supplementary Fig. 1c–e). The common developmental onset of all three gaze-stabilizing eye movement components coincided with a major step in the maturation of the rhythmically alternating spinal locomotor output, which changes from single-spikes in Vrs at stage 37/38 to spike bursts at stage 42 (Supplementary Fig. 1a)[23]. In contrast, larvae between stage 40 and stage 48 were unable to produce an angular VOR (aVOR), due to the incomplete or insufficiently-sized semicircular canals (Fig. 1d)[20].

### Joint maturation of locomotor-induced ocular motor behavior and angular vestibulo-ocular reflexes. Spino-ocular motor coupling experiences an extensive spatio-temporal plasticity during metamorphosis, culminating during the emergence of appendicular and concurrent loss of axial muscle-based locomotor strategy in post-metamorphic froglets[10,24]. To avoid interference with plasticity processes related to this latter transition of the body shape at stage 59, the evaluation of the developmental plasticity in spino-ocular motor coupling was therefore restricted to larvae prior to this stage (Fig. 2a). At stage 49, ascending spinal locomotor efference copies, in the absence of motion-related visuo-vestibular sensory inputs, elicited phase-coupled oscillatory eye movements with relatively invariable magnitudes of 5–7° (Fig. 2a₁–b₁) independent of the tail bending amplitudes ($R = 0.05 \pm 0.05$, $r^2 = 0.02 \pm 0.02$, $N = 8$; Fig. 2b₁,₂; Supplementary Fig. 2g). Consequently, the gain for locomotor efference copy-induced eye movements was relatively high for small tail excursions ($0.74 \pm 0.10$ for left-right tail deflections of ~10°, $N = 8$; Fig. 2b₃, left panel) but decreased gradually for larger tail oscillations (Fig. 2b₃; Supplementary Fig. 2a). In contrast, at stage 58, eye and tail movement amplitudes were more closely correlated ($R = 0.56 \pm 0.11$; $r^2 = 0.46 \pm 0.07$, $N = 8$; Fig. 2b₁,₂ and Supplementary Fig. 2g) with a lower but invariant gain of ~0.4, independent of tail excursion magnitudes (Fig. 2b₃, left panel; Supplementary Fig. 2b). The spino-ocular motor coupling revealed an average phase lag *re* deflection of the rostral tail region of 0–30° (Fig. 2b₃, right panel) that was unrelated to tail excursion magnitude and independent of age. The relative timing

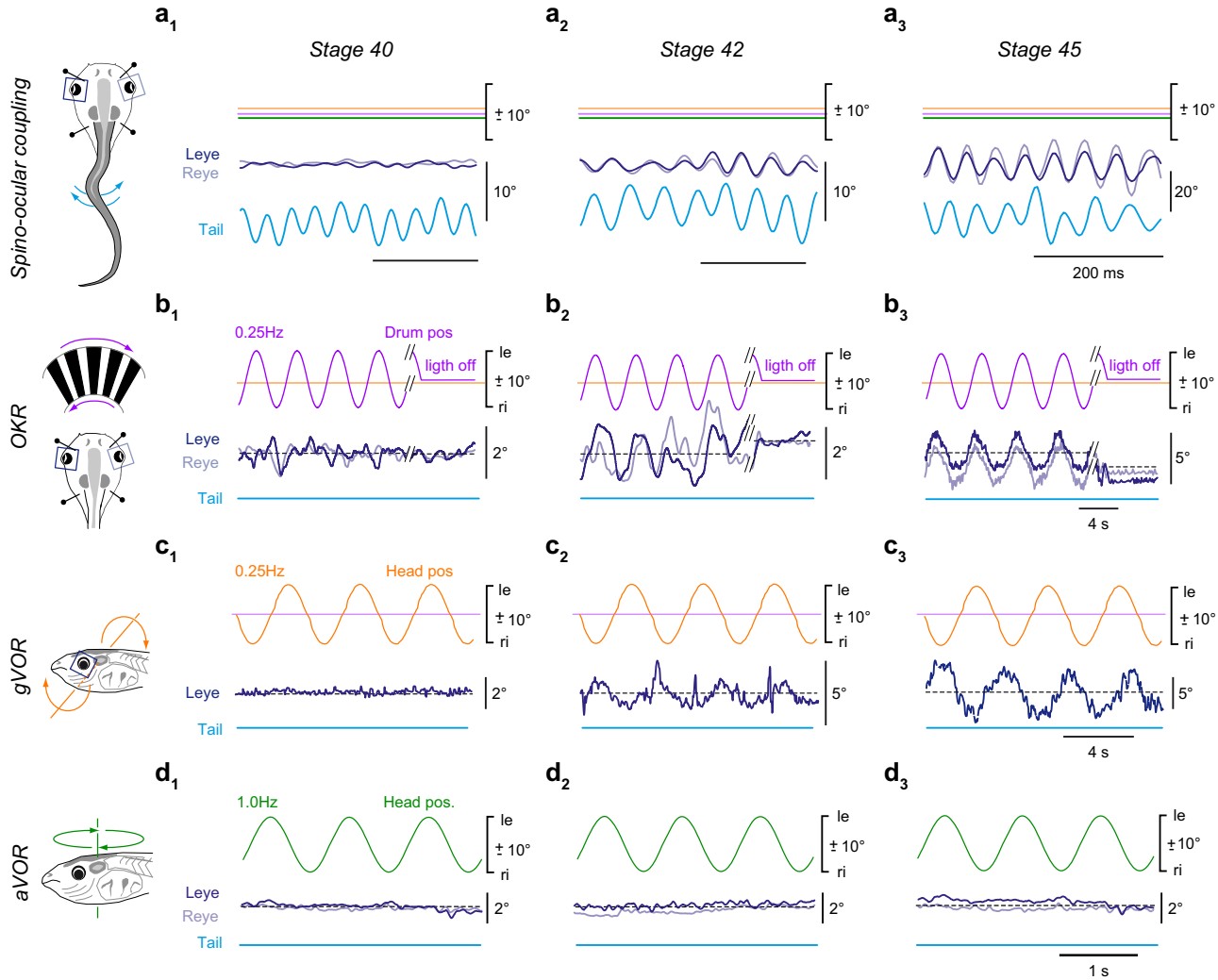

**Fig. 1 Ontogeny of spino-ocular, visual motion-, otolith organ- and semicircular canal-driven eye movements.** Gaze-stabilizing movements of the left (Leye) and right eye (Reye) during spino-ocular motor coupling (**a**), optokinetic reflex (OKR, **b**) gravitoinertial (gVOR, **c**) and horizontal angular vestibulo-ocular reflex (aVOR, **d**) at stage 40 (**a₁, b₁, c₁, d₁**), stage 42 (**a₂, b₂, c₂, d₂**) and stage 45 (**a₃, b₃, c₃, d₃**). Swim episodes occurred spontaneously; OKR and gVOR were elicited by sinusoidal rotation of black/white vertical stripes or downward-upward head roll motion at 0.25 Hz with positional excursions (pos) of ±10° during horizontal head rotation at 1 Hz with positional excursions of ±10° an aVOR was absent at developmental stages 40–45. le, ri, leftward, rightward movement.

of the responses was rather variable at stage 49 but became considerably more homogeneous in older larvae (polar plot inset in Fig. 2b₃ and Supplementary Fig. 2d, e, compare vector length of the mean phase relation).

The developmental period between stage 49 and 58 in *Xenopus* tadpoles is generally characterized by a considerable improvement of the performance of the horizontal aVOR (Fig. 2c, d). First noticeable compensatory eye movements during horizontal head/body rotation were observed as early as stage 49–50 (Fig. 2c₁; N = 10), although with a gain below 0.2, independent of stimulus strength (Fig. 2d₁). This relatively late aVOR onset, compared to gVOR and OKR onsets is related to the necessity of semicircular canals to acquire sufficiently large duct diameters to become operational, which in *Xenopus* occurs shortly before stage 49–50[21]. Thereafter, aVOR gains became significantly increased by stage 52 (N = 6), especially during large amplitude head rotations (1 Hz, ± 15–25°; Fig. 2d₁, e) with further enhancement, although at a slower rate, until stage 58 (Fig. 2d, e; N = 5). The aVOR phase relation during sinusoidal rotation remained stable between stage 49 and stage 58, with small phase leads of the response *re* stimulus (~0–30°) at 1 Hz (Fig. 2d₂).

Interestingly, the aVOR gain was uncorrelated with the tail oscillation amplitude across the developmental period from stage 49 to stage 58 (Supplementary Fig. 2f).

Collectively, these findings suggest that the implementation and refinement of locomotor efference copy-driven eye movements are closely intertwined with the developmental maturation of visuo- and vestibulo-ocular motor reflexes. Accordingly, spino-ocular motor coupling co-emerges together with visuo- and otolith organ-driven ocular motor responses at stage 42, followed by a progressive improvement in performance with a common time course until stage 49 (Fig. 1, Supplementary Fig. 1). Thereafter, further enhancement of locomotor efference copy-driven eye movements coincides with the maturation of the horizontal aVOR, with a potentially cross-functional interaction of the two ocular motor behaviors (Fig. 2e).

**Impact of semicircular canals on the maturation of ocular motor behavior.** The potential impact of a gradually improving aVOR performance on the maturation of locomotion and

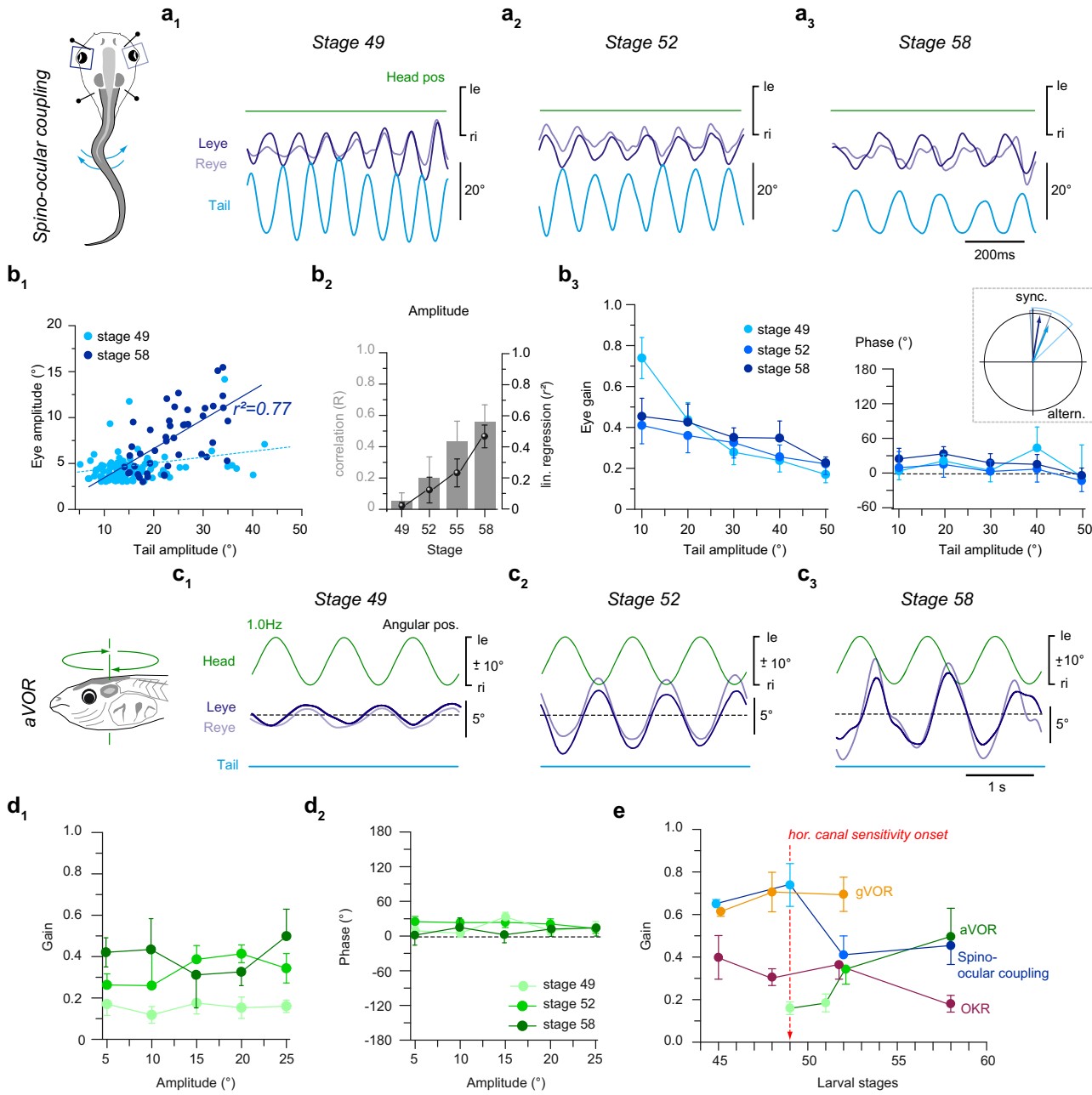

**Fig. 2 Developmental improvement of swim-related predictive ocular motor behavior and horizontal angular vestibulo-ocular reflex performance.**
**a** Movement of the left (Leye) and right eye (Reye), triggered by spino-ocular motor coupling (pictogram) at stage 49 (**a₁**), 52 (**a₂**) and 58 (**a₃**) during undulatory tail movements. **b** Quantification of spino-ocular motor coupling parameters, presented as scatter plot of eye motion amplitudes relative to tail motion amplitudes (**b₁**), histogram (gray bars) of the correlation coefficient (R) and linear (lin.) regressions (r², black spheres) between stage 49 and 58 (**b₂**, stage 49: N = 8, stage 52: N = 6, stage 55: N = 7, stage 58: N = 9), linearity analysis of the eye position gain (left in **b₃**, stage 49: N = 8, stage 52: N = 5, stage 58: N = 7) and phase (right in **b₃**, stage 49: N = 10, stage 52: N = 4, stage 58: N = 6) as function of the tail undulation amplitude for three representative stages (see color-code); data are presented as mean values ± SEM. The inset illustrates a polar plot of the average phase relation (mean ± 95% confidence intervals; detailed polar plots shown in Supplementary Fig. 2d, e) at stage 49 and stage 58. **c** Angular vestibulo-ocular reflex (aVOR), elicited by sinusoidal horizontal head rotation (pictogram) at 1 Hz and ±10° positional excursions at stage 49 (**c₁**), 52 (**c₂**) and 58 (**c₃**) in the absence of swim-related tail undulations (light blue line) in darkness. **d** Linearity analysis of the aVOR gain (**d₁**) and phase (**d₂**) plotted as function of the head motion amplitude for three representative stages (see color-code; stage 49: N = 4, stage 52: N = 5, stage 58: N = 5) during sinusoidal rotations at 1 Hz (mean ± SEM). **e** Summary plot of the eye position gain (mean ± SEM) as index for the proficiency of spino-ocular motor coupling, aVOR, OKR and gVOR during larval development from stage 45 to stage 58. Numbers of animals (N) used in this plot were equivalent to the numbers of animals employed in Supplementary Figs. 1d, e and 2b₃–d₁ at different developmental stages. le, ri, leftward, rightward movement. Source data are provided as a Source Data file.

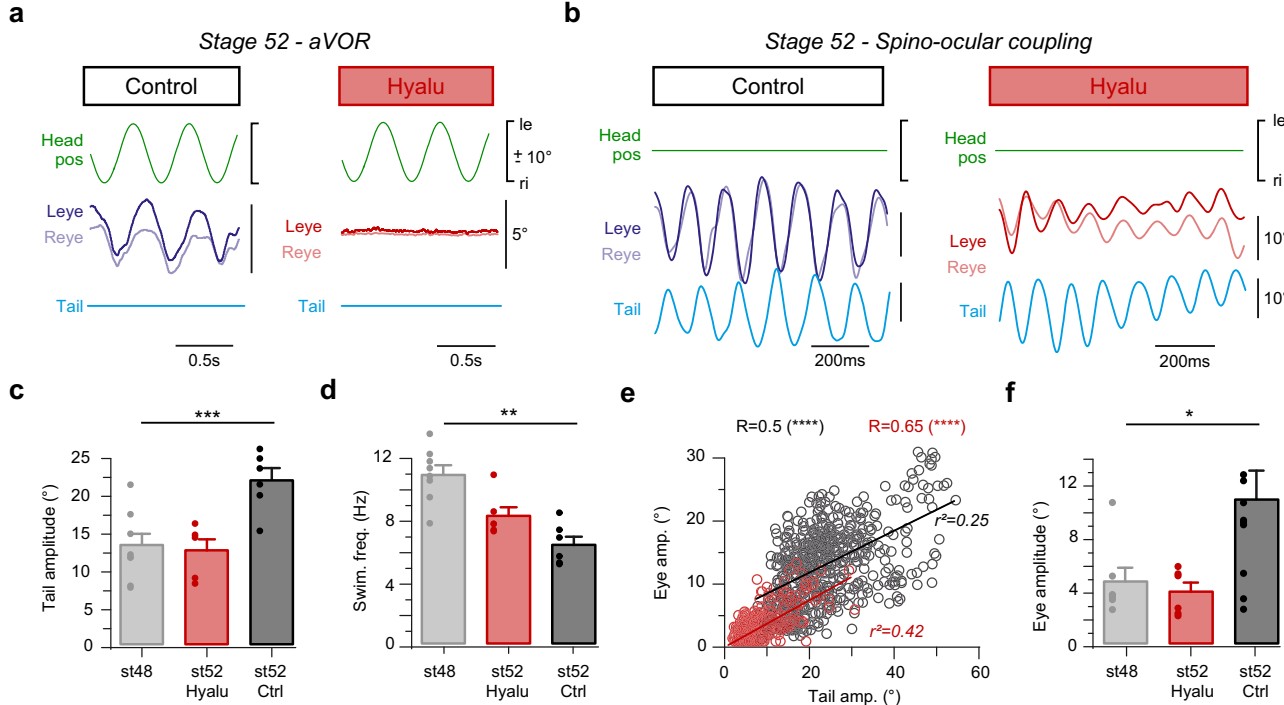

**Fig. 3 Semicircular canal deficiency prevents maturation of spino-ocular motor coupling. a** Movements of the left (Leye) and right eye (Reye) indicative for the angular vestibulo-ocular reflex (aVOR) evoked by horizontal sinusoidal head rotation at 1 Hz and ±10° positional (pos) excursion in the absence of swimming (**a**) and for an episode of spino-ocular motor coupling during tail undulations in a head-stationary preparation (**b**); recordings were obtained from a stage 52 control (left column in (**a**), (**b**)) and an age-matched semicircular canal-deficient tadpole (right column in (**a**), (**b**)), respectively; semicircular canal formation was prevented by bilateral intra-otic injections of hyaluronidase (Hyalu) at stage 44. Histogram of swim-related tail undulation amplitude (**c**) and frequency (**d**) during swimming of stage 48 controls (light gray bars, $N = 8$), hyaluronidase-treated stage 52 animals (red bars, $N = 6$) and stage 52 controls (dark gray bars, $N = 7$); statistical significance was evaluated by the Kruskal-Wallis test, $p = 0.002$ and $p = 0.0003$, respectively, for histograms (**c**) and (**d**). Data are presented as mean values ± SEM (**e**) Quantification of locomotion-induced ocular motor coupling, depicted as scatterplot of eye position amplitudes relative to tail motion amplitudes and linear (lin.) regressions ($r^2$) of stage 52 controls (black circles, $n = 560$ cycles) and hyaluronidase-treated age-matched animals (red circles, $n = 375$ cycles). Two-tailed $p$ values were calculated to estimate the Pearson coefficient correlation (R) significance and were less than 0.0001. **f** Histogram of average eye position amplitudes during spino-ocular motor coupling of stage 48 controls (light gray bars, $N = 8$), hyaluronidase-treated stage 52 animals (red bars, $N = 6$) and stage 52 controls (dark gray bars, $N = 7$); statistical significance of the Kruskal-Wallis test, $p = 0.0209$. Data are presented as mean values ± SEM. Statistical significance is indicated as *** $p < 0.001$, ** $p < 0.01$, * $p < 0.05$ in (**c**–**f**); le, ri, leftward, rightward movement. Source data are provided as a Source Data file.

associated spino-ocular motor coupling was elucidated in semicircular canal-deficient stage 52 larvae. These animals were obtained following bilateral injection of hyaluronidase into both otic capsules at stage 44 prior to inner ear duct completion[25]. This experimental perturbation at early larval stages caused a failure to form semicircular canals on both sides and consequently disabled the aVOR at later stages[26]. In compliance with the absence of functional semicircular canals, imposed horizontal head rotation at stage 52 failed to elicit compensatory eye movements (Fig. 3a). Despite the absence of an aVOR, hyaluronidase-treated animals were able to produce undulatory swimming and locomotor efference copy-induced eye movements (Fig. 3b). However, the performance of both swim strength and associated ocular motor behavior was inferior to that of age-matched controls and more comparable to that of younger larvae (Fig. 3c–f). Undulatory tail movements in these semicircular canal-deficient stage 52 larvae were characterized by small amplitudes and high frequencies (Fig. 3c, d), reminiscent of those, observed in younger, stage 48 controls prior to the aVOR onset[27]. These reduced tail movement amplitudes were accompanied by small locomotor-induced eye movements in these vestibular deficient animals (Fig. 3e, f). This suggests that semicircular canal signals are not only required for a functional aVOR, but obviously also for the maturation of the locomotor capacity[22,23,28] and associated spino-ocular motor

coupling with stage-specific dynamic profiles. Direct causality under these circumstances is difficult to assess (see discussion), however, a direct impact of the aVOR on the maturation of locomotor efference copy-driven eye movements represents one of the potential scenarios.

**Progressive developmental interactions between semicircular canal- and locomotion-induced eye movements**. The interaction between semicircular canal- and locomotor efference copy-driven eye movements was further evaluated by calculating the impact of horizontal head rotations on concurrent spino-ocular motor performance during undulatory swimming between developmental stages 48 and 58. Discrimination of vestibular- and locomotor efference copy-driven response components in resultant eye movements was achieved by spectral analyses, which required activation of the two components at different frequencies, respectively. Accordingly, passive head rotation was applied at 0.5–2 Hz (± 10° positional excursions), known to elicit a robust aVOR in larvae older than stage 49[21]. This stimulus frequency was sufficiently distant from swim-related tail beat oscillations that occurred within an age-dependent range of 6–15 Hz[27].

At stage 48, oscillatory eye movements during swim-related tail undulations and concurrent horizontal sinusoidal head rotation

were robust (Fig. 4a$_1$). However, noticeable 1 Hz vestibular motion stimulus-related eye movement components were clearly absent as indicated by the lack of the respective spectral component (see absence of red band at 1 Hz in the spectrogram in Fig. 4b$_1$ and of peak at 1 Hz in the periodogram in Fig. 4c$_1$). This lack complies with the known failure to elicit an aVOR at this developmental stage[21]. The eye movement frequency component at stage 48 therefore corresponds exclusively to the swim frequency (~12 Hz), indicated by the red band in the spectrogram and by the respective peak in the periodogram (Fig. 4b$_1$, c$_1$). As expected, at stage 52, eye movements during tail undulations and simultaneous sinusoidal horizontal head rotation were also rather robust (Fig. 4a$_2$). Spectral analysis demonstrated, however, that the eye movements contained two clearly discernable temporal components that coincided with the vestibular stimulus frequency (1 Hz) and the tail beat oscillation (~6 Hz), indicated in the typical example by the red bands in the spectrogram (Fig. 4b$_2$) and the corresponding peaks in the periodogram (Fig. 4c$_2$).

Respective spectral analyses of eye movements during swimming and concurrent head rotation in larvae between stage 45 and 58 suggests a progressive reconfiguration of sensory-motor and predictive spino-ocular motor response components (Fig. 4d, e). Horizontal head rotation-induced eye movement components experienced a gradual, yet significantly increasing contribution to the net motor outcome (Kruskal-Wallis test, $p < 0.001$, $N = 17$) after the onset of semicircular canal function at stage 49, culminating at stage 58, just prior to the metamorphic switch from tail- to limb-based locomotion (Fig. 4d, e). Concomitantly, the influence of CPG-induced spino-ocular motor components on eye movements demonstrated a progressive diminution during this larval period (decrease of the locomotor power peak, Kruskal-Wallis test, $p < 0.0001$; Fig. 4f). Possible underlying plasticity mechanisms include an increasing efficacy of peripheral and central processing of semicircular canal signals. This obviously has a direct impact on extraocular motoneuronal activity as well as on spinal central pattern generator performance.

The increasing influence of angular vestibular motion signals on the ocular motor output during swimming was confirmed by recordings of the spike discharge of the left *lateral rectus* (LLR) nerve, a more sensitive indicator for signal integration, along with the motion of the right eye in semi-intact preparations at stage 52 (Supplementary Fig. 3a, b). Spectral analyses of the eye movements and LLR nerve discharge revealed two, distinct frequency components (Supplementary Fig. 3c$_1$, c$_2$, red bands in Supplementary Fig. 3d and peaks in Supplementary Fig. 3e), which were of semicircular canal and spinal locomotor CPG origin, respectively (Supplementary Fig. 3e). These two response components jointly produced after their fusion compensatory eye movements. Most interestingly, however, the cyclic spinal Vr discharge was broadly tuned, producing a spindle-shaped envelope with the timing of the head rotation frequency (Supplementary Fig. 3b). This finding provides suggestive evidence that horizontal semicircular canal signals modulate the activity of spinal CPG circuits with direct implications for the strength of locomotor commands and resultant ascending copies thereof, designated as premotor constituent of ocular motor signals.

**Developmental plasticity in the integration of vestibular and locomotor signals into gaze-stabilizing ocular motor responses.** Eye movements, produced during locomotor activity and simultaneous horizontal head rotation appeared to be governed by cross-modal interactions that principally affected two ocular motion parameters (see Fig. 5 and Supplementary Fig. 4) either

separately or jointly as illustrated in the example obtained from a stage 52 larva shown in Fig. 5a$_1$–b$_1$. The two parameters are: eye motion magnitude, which is the amplitude of efference copy-induced oscillatory movements and eye position eccentricity, which indicates the medio-lateral orientation of the eye in the orbit relative to the longitudinal axis of the head (see Fig. 5b$_{1-3}$). Accordingly, both the ocular motion magnitude and the eccentricity of swim cycle-related eye movements were modulated during sinusoidal head rotation (Fig. 5a, b) with an alternating enhancement and attenuation depending on the rotation half cycle (green dots in Fig. 5b$_2$ for magnitude; magenta squares in Fig. 5b$_3$ for eccentricity). Variations of the swim-related ocular motion magnitude (Δ magnitude, Fig. 5b$_2$) were in the range of 1–11° (2–8° for the trace shown in Fig. 5a$_1$, see Supplementary Fig. 4a$_1$) between the largest and smallest eye excursions ($N = 10$). This variation appeared to be independent of particular swim parameters (swim frequency in Fig. 5c$_1$, tail beat amplitude in Supplementary Fig. 4b$_1$) as well as of the vestibular stimulus frequency at least across the explored range from 0.5–2 Hz (Fig. 5d$_1$ and Supplementary Fig. 4c$_1$).

The observed variations of the swim-related ocular eccentricity (Δ eccentricity, Fig. 5b$_3$) were in the range of 0–15° (2–9° for the trace shown in Fig. 5a$_1$; see Supplementary Fig. 4a$_2$) between the largest and smallest eye eccentricity angles ($N = 10$). In contrast to the magnitude variation, the extent to which the vestibular-induced ocular eccentricity was varied was noticeably correlated with the swim frequency, with a considerable attenuation at high tail beat frequencies (Fig. 5c$_2$; see Supplementary Fig. 4a$_2$). However, the variation in eye positional eccentricity remained uncorrelated with respect to the swim tail amplitude (Supplementary Fig. 4b$_2$). However, eye positional eccentricity was differentially, although weakly influenced by the vestibular stimulus with higher motion stimulus frequencies (i.e., 2 Hz) causing smaller positional alterations (Fig. 5d$_2$; Supplementary Fig. 4c$_2$). Interestingly, this attenuating influence of locomotor efference copies on the vestibular-driven eye eccentricity became more and more attenuated with weaker swimming (lower tail undulation frequency) as occasionally observed when the swim strength decreased spontaneously during a given swimming episode in a particular tadpole (Fig. 5a).

The modulation of CPG efference copy-related ocular motor responses by horizontal head rotation during swim episodes was confirmed by the spike discharge of the LLR motor nerve in semi-intact preparations at stage 52 (Supplementary Fig. 4e, f). Spike burst amplitudes (light red dots in Supplementary Fig. 4e, f) as well as baseline spike activity (red square in Supplementary Fig. 4e, f) in the swim-related LLR discharge were robustly modulated during the head rotation with a profile that was phase-timed to the imposed motion (Supplementary Fig. 4e$_{2,3}$, f$_{2,3}$). The vestibular-induced variation of these two swim-related LLR spike discharge parameters was compatible with variations of eye motion magnitude and eye position eccentricity during simultaneous swim episodes and horizontal head rotation. The variation of LLR burst amplitude, however, could occur independently (Supplementary Fig. 4e) or in combination with the variation of the LLR baseline activity (Supplementary Fig. 4f).

The vestibular-induced modulation of the swim-related ocular eccentricity increased significantly during development (Δ eccentricity $= 3.78 \pm 1.04°$ at stage 49, $3.89 \pm 0.47°$ at stage 52, $12.09 \pm 3.17°$ at stage 58; Kruskal-Wallis test $p < 0.05$; Fig. 5e$_2$). In contrast, the modulation of the swim-related ocular motion magnitude remained relatively constant across development between stage 49 and stage 58 (Δ magnitude $= 6.23 \pm 1.27°$ at stage 49, $5.12 \pm 0.62°$ at stage 52, $8.05 \pm 1.42°$ at stage 58; Kruskal-Wallis test $p = 0.1829$; Fig. 5e$_1$). Despite the clear evidence for an age-related progressive increase in the vestibular-induced

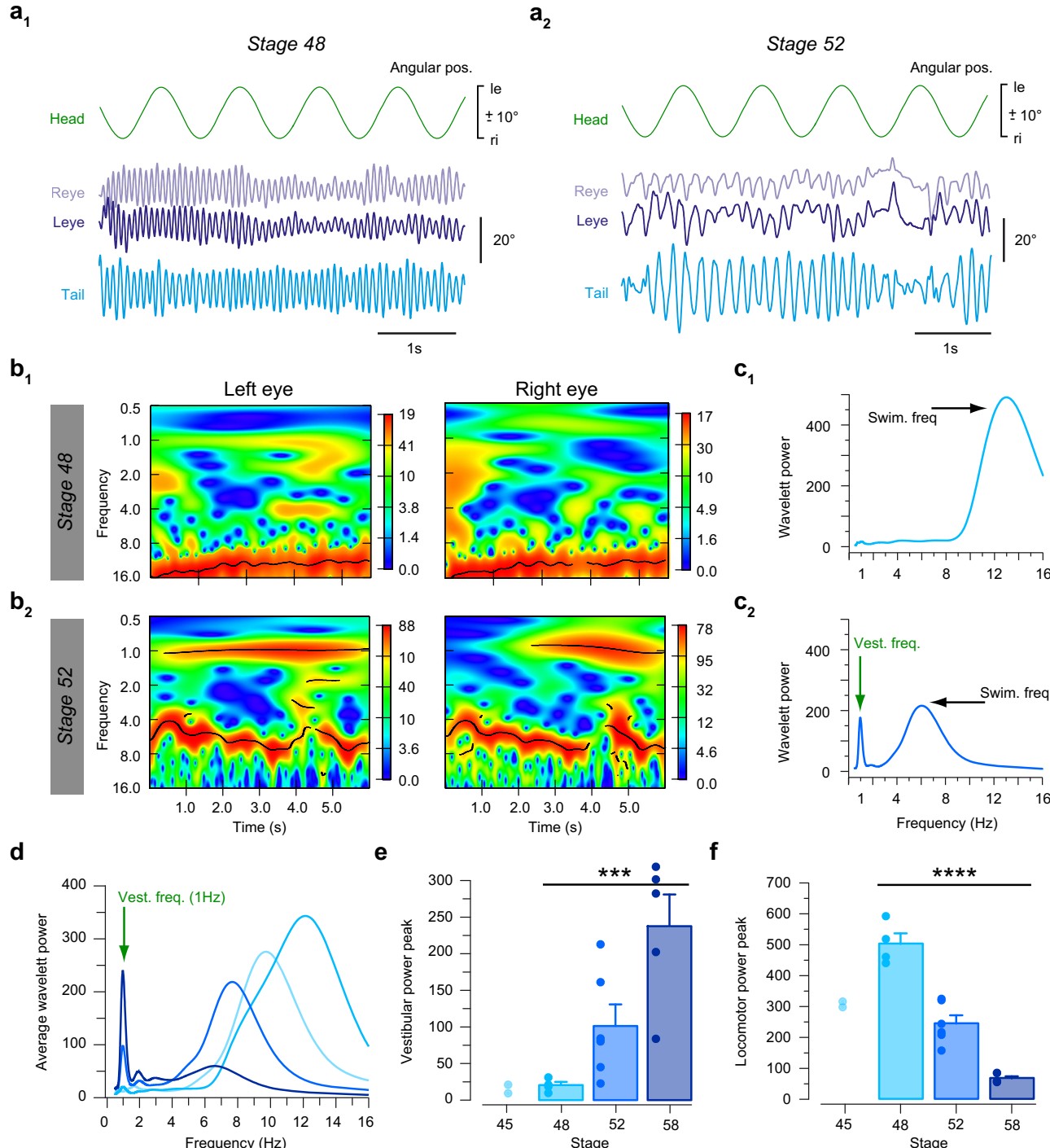

**Fig. 4 Stage-specific integration of semicircular canal and spino-ocular motor signals during locomotion.** Representative recordings (**a**) and spectrograms (**b**) depicting the power spectrum of movement frequencies of the left (Leye) and right eye (Reye) at stage 48 (**a₁, b₁**) and stage 52 (**a₂, b₂**) during swim-related tail undulations (blue trace in **a₁,₂**) and concurrent horizontal head rotation at 1 Hz with positional excursions of ± 10° (green traces in **a₁,₂**); the occurrence of each frequency within the eye motion profile from 0.5–16 Hz is indicated by the color gradient from blue (rare) to red (often); black lines refer to significantly different wavelet power values within the spectrum. Color scales, blue to red, at the side of each spectrogram represents the amplitude of wavelet components, low to high respectively, relative to the occurrence of each frequency. **c** Periodograms of two representative recordings at stage 48 (**c₁**) and stage 52 (**c₂**); note that eye movements at stage 48 consist of a single principal frequency that coincides with the tail undulations (Swim. freq. ~13 Hz), while those at stage 52 contain two clearly discernible frequencies representing tail undulations (Swim. freq. ~6 Hz) and horizontal head rotation (Vest. freq. 1 Hz). Average periodogram of eye movements (**d**) during combined swim-related tail undulations and head rotations at 1 Hz at selected stages (color-coded); note the developmental changes of the frequency component related to the vestibular stimulus at 1 Hz (**e**) and the swimming frequency (**f**), summarized in the histograms (mean values ± SEM) of average wavelet power amplitudes (**e, f**) at stage 48 ($N = 4$), 52 ($N = 6$) and 58 ($N = 5$); statistical significance using the Kruskal-Wallis test was performed between these stages. Exact $p$ values calculated were equal to 0.001 for histogram (**e**) and less than 0.0001 for histogram (**f**). Note that values of stage 45 ($N = 2$) were not used in descriptive statistics and statistical test due to the small animal number. Statistical significance is indicated as **** $p < 0.0001$ and *** $p < 0.001$ in (**e, f**); le, ri, leftward, rightward movement. Source data are provided as a Source Data file.

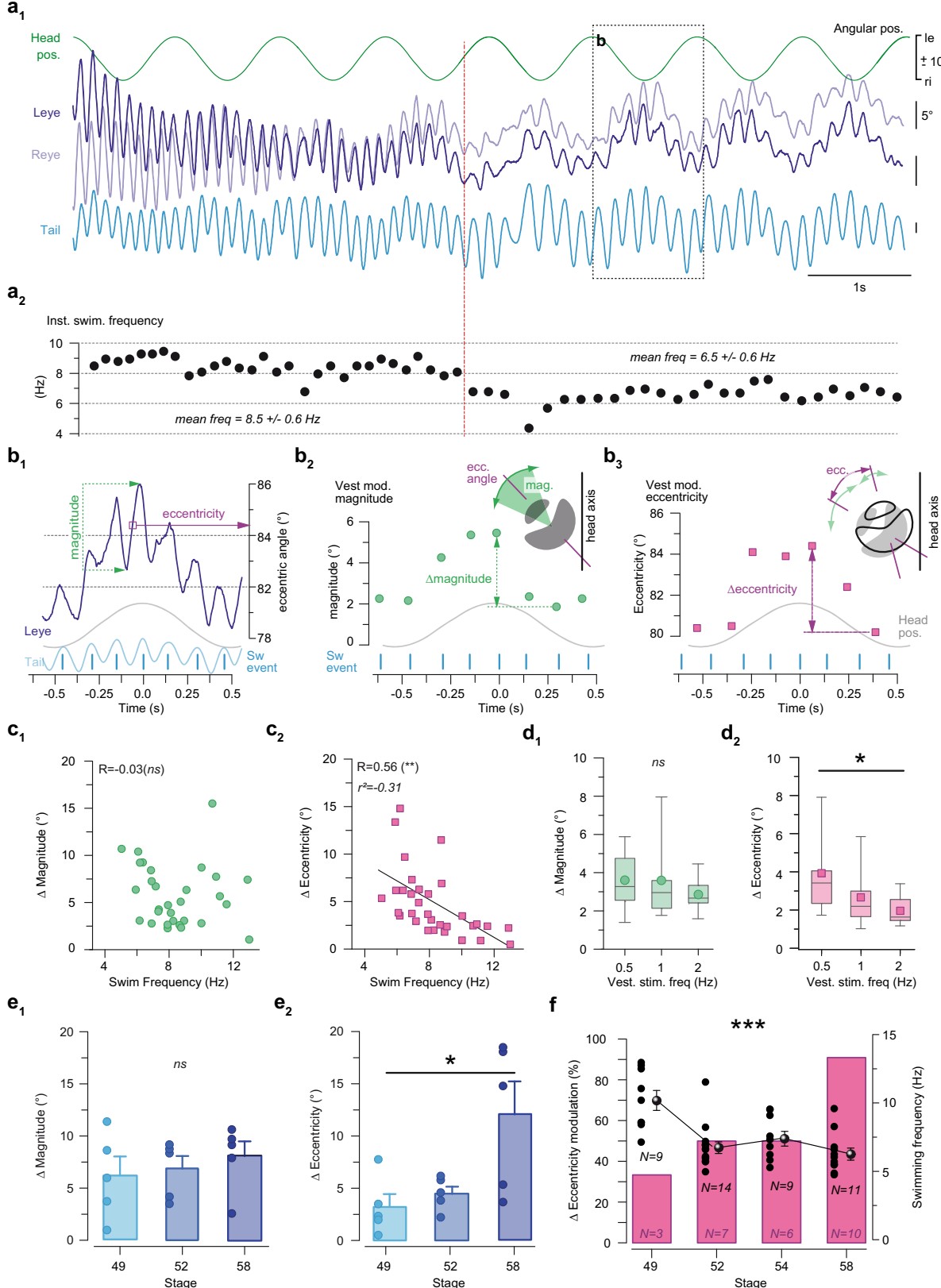

modulation of the ocular eccentricity during swimming (Fig. 5e₂), larval age might only be an indirect classifier. As demonstrated earlier[9,27], the frequency of swim-related tail oscillations decreases progressively during larval development (black dots and spheres in Fig. 5f). Thus, when swim frequency is used as parameter, vestibular-induced modulation of the ocular

eccentricity occurred predominantly at lower tail undulatory frequencies (~6 Hz, Fig. 5f). Such low tail undulation frequencies are characteristic for the swimming of older larvae (see Fig. 4d, f), while exclusive variations of the ocular motion magnitudes were found, whenever swimming occurred at higher frequencies, which are more typical for stage 49 larvae (~10 Hz; Fig. 5f).

**Fig. 5 Vestibulo-ocular and spino-ocular motor signal integration depends on the swim frequency. a** Movements of the left (Leye) and right eye (Reye) during horizontal sinusoidal head rotation (green) and swim-related tail undulations (blue) at stage 52 (**a₁**); note that the spontaneously reduced swim frequency within the illustrated episode (marked by the red dashed line in **a₂**) produced two different types of resultant ocular motion. **b** Representative example of ocular motion of the left eye during a single horizontal head rotation cycle and concurrent tail undulations (**b₁**), depicting the variation in eye motion magnitude (**b₂**) and change of eye position eccentricity (= angular position of the eye relative to the longitudinal axis of the head; **b₃**). **c** Quantification of ocular motion parameters as scatter plots of Δ motion magnitude (**c₁**) and Δ position eccentricity (**c₂**) as function of the swimming frequency ($n = 32$ swimming episodes from $N = 15$ animals at different developmental stages). Two-tailed $p$ values were calculated to estimate the Pearson coefficient correlation (R) significance ($p = 0.3023$ for the Δ motion magnitude plot and $p = 0.0023$ for the Δ motion eccentricity plot). **d** Box plots of Δ motion magnitude (**d₁**) and Δ position eccentricity (**d₂**) as function of the vestibular stimulus frequency at 0.5 Hz ($n = 10$ swimming episodes), 1 Hz ($n = 13$ swimming episodes) and 2 Hz ($n = 14$ swimming episodes) from $N = 4$ animals. Upper and lower error bars of box plots represent, respectively, maximum and minimum values. Bounds of box and center lines represent, respectively, the values of 50% of the central region (75 and 25% percentile) and the median values. Superimposed full circles/squares represent mean values. $p$-values, calculated using the Kruskal-Wallis test were equal to 0.6754 and 0.0423 respectively for box plots (**d₁**) and (**d₂**). **e** Histogram of Δ magnitude (**e₁**) and Δ eccentricity (**e₂**) variations during spino-ocular motor coupling at stage 49 ($N = 5$), stage 52 ($N = 5$), and stage 58 ($N = 5$). The Kruskal-Wallis test was performed to test the significance and resulted in $p = 0.1829$ in **e₁** and $p = 0.0343$ in **e₂**. **f** Relative occurrence of eccentricity modulation during ocular motion (left axis) and average swim frequency (black dots and spheres, right axis) between stage 49 and stage 58 (Kruskal-Wallis test, $p = 0.0007$). Data are presented as mean values ± SEM and statistical significance is indicated as *** $p < 0.001$, ** $p < 0.01$, * $p < 0.05$, ns non-significant in (**c-f**). Source data are provided as a Source Data file.

The obvious swim frequency-dependent effect of head rotation on either swim-related eye motion magnitude or eye position eccentricity therefore suggests that the interaction between locomotor efference copies and vestibular sensory signals could involve two different vestibular pathways that become gradually altered in their computational efficacy during larval development. Accordingly, at stage 49, horizontal semicircular canal signals modify spinal CPG activity, through vestibulo-spinal pathways, while the VOR-related motor output is suppressed[8] at high swimming frequencies, preferentially expressed in these young tadpoles. As a result, semicircular canal activation indirectly modulates locomotor efference copy signals by modulating the CPG activity, which in turn induces exclusively a modulation of the swim-related ocular motion magnitude. In older animals, at stage 58, VOR gating[8] by locomotor efference copies disappears at the typically lower swim frequencies and instead is replaced by a summation of intrinsic CPG and sensory-motor components in the process of synthesizing the final ocular motor commands. This causes a direct VOR-induced variation of the swim-related ocular positional eccentricity in addition to the more indirect variation of the eye motion magnitude. Key to this developmental switch in this view is a gradually reduced efficacy of the locomotor program, visible as slower tail rhythm during the metamorphic progression with concurrent vanishing influences of locomotor efference copies on vestibular sensory components in the ocular motor output.

This hypothesis was tested by exploiting a computational model of vestibulo-ocular, vestibulo-spinal, spinal CPG[29] and spino-ocular motor pathways[8] collectively involved in the generation of ocular motor responses during combined head rotations and locomotor activity (Fig. 6a). Several currents were generated either separately or simultaneously to concomitantly drive the discharge activity of abducens motoneurons (Abd MN in orange; Fig. 6a). To elicit rhythmic motor patterns, a constant current (see methods) was injected for 5 s into the CPG neurons (dIN) responsible for rhythmic spinal motoneuron activity (sp MN in blue; Fig. 6a). Consequently, current injection at an intensity of 0.15 nA produced a coupled burst discharge at 7 Hz in both sp MN and Abd MN (Fig. 6b and Supplementary Fig. 5a) comparable to locomotor efference copy-induced spino-ocular motor coupling observed in real biological preparations[8]. The impact of a horizontal head rotation was simulated by an injection of a sinusoidal current at 1 Hz into vestibulo-ocular neurons (VO in green; Fig. 6a), producing a phase-coupled discharge in Abd MN (Fig. 6c and Supplementary Fig. 5b) comparable to a classical VOR response[20,30]. Sinusoidal currents, injected into vestibulo-spinal

neurons (VS in green; Fig. 6a) also produced a distinct activity in sp MNs similar in outcome to a vestibulo-spinal reflex[31]. Current injections > 0.16 nA into dIN CPG neurons (causing swimming at an undulatory frequency > 8 Hz; Fig. 6d₁, e) produced a resultant inhibition of VO neurons (Fig. 6a) that were concurrently driven by a 1 Hz sinewave current, however, without affecting the VS neuron discharge (see "Methods").

Consequently, based on the model, sinusoidal current injections at 1 Hz into both VO and VS neurons and constant current injections of 0.16 nA into both dIN neurons (Fig. 6e) produced a coupled burst activity in Abd MN that matched the 8 Hz rhythmic discharge of sp MN (Fig. 6d₁ and Supplementary Fig. 5c). In that case the swim-related, integrated Abd MN burst amplitude (see inset scheme in Fig. 6b) was modulated with the timing of the sinusoidal current injected at 1 Hz into VO/VS neurons. The smaller and the larger burst amplitude were phase-related to the 1 Hz sinusoidal stimulus (Fig. 6d₁ and Supplementary Fig. 5c), independent of the intensity of the current injected into dIN (Fig. 6f₁). Interestingly, the integrated Abd MN burst amplitude was correlated with its discharge frequency. However, the Abd MN baseline discharge (see inset in Fig. 6b) was only barely modulated in this configuration as well as for even higher current intensities that were injected into dIN (Fig. 6f₂).

Similar current injections at 1 Hz into both VO and VS neurons along with current injections at a magnitude of 0.13 nA into dIN CPG neurons produced a burst activity in Abd MN that was coupled to the 6 Hz rhythmic discharge of the sp MN (Fig. 6d₂ and Supplementary Fig. 5d). With these parameters, the swim-related, integrated Abd MN burst amplitude and baseline spike activity revealed a modulation that was in phase with the 1 Hz sinusoidal activity of the VO/VS neurons. Thereby, the VO neuron-induced variation of the Abd MN burst amplitude appeared to be independent of the intensity of the current injected into dIN (Fig. 6f₁). In contrast, the VO neuron-induced variation of the Abd MN baseline burst activity decreased linearly with increasing current amplitudes injected into dIN (Fig. 6f₂). Therefore, as an essential outcome of the modeling approach, the swim frequency, determined by the injected amount of current into the dIN CPG neurons, was the decisive factor for the vestibular modulation of the swim-related Abd MN baseline burst discharge, likely through an enhancement of the inhibitory current in VO neurons above a frequency of 8 Hz (dIN injected current > 0.16 nA; Fig. 6e). The vestibular modulation of the swim-related Abd MN burst amplitude therefore represents a residual effect of the impact of VS neuronal activity on the sp MN/EC neuron (efference copy neuron), independent of the swim frequency.

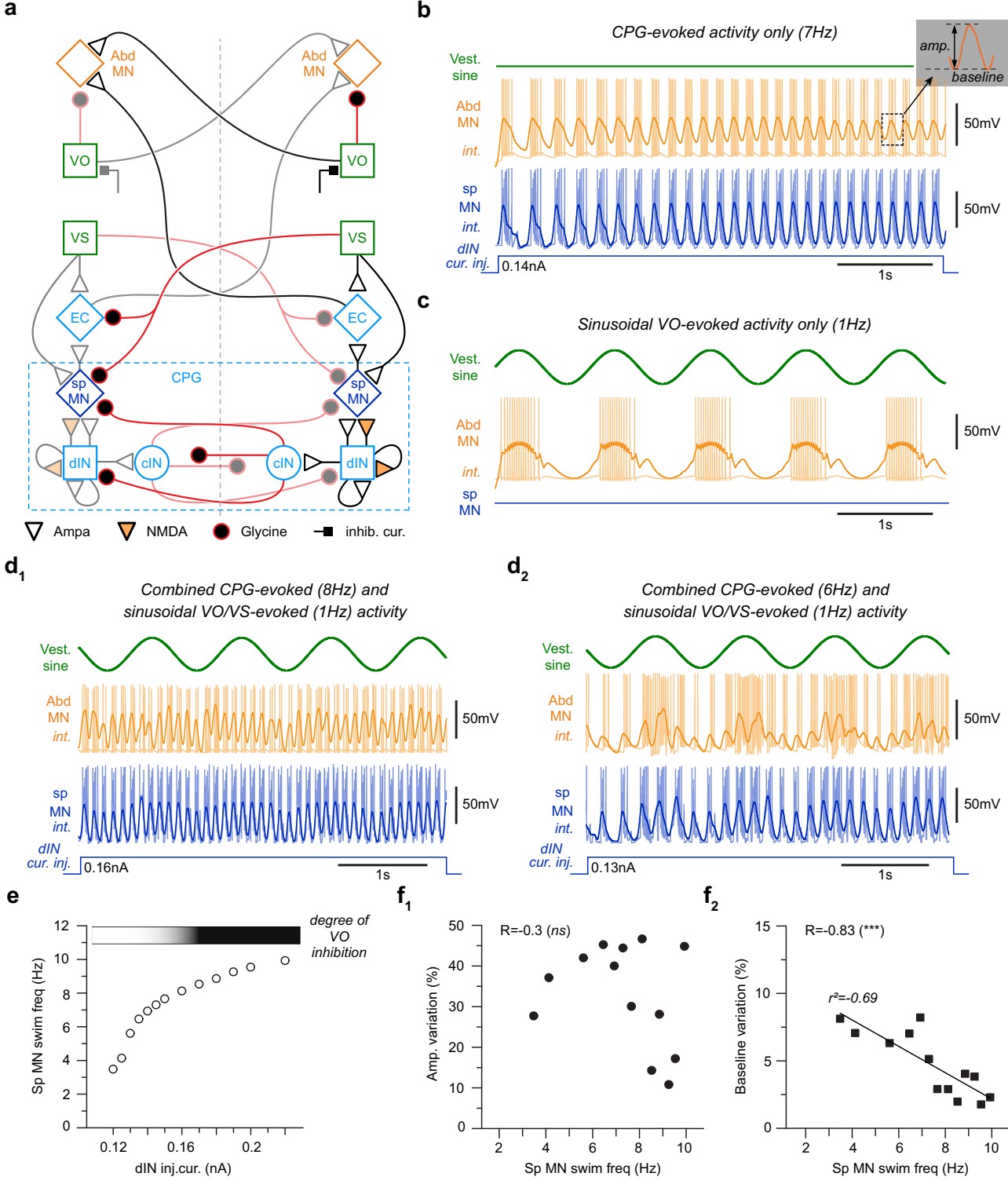

Based on the outcome, the computational modeling was able to reproduce the different ocular motor activity patterns generated during combined head rotation and swim episodes described in semi-intact larval preparations (Fig. 5). These modeling results confirmed the differential impact of the swim frequency on the vestibular-driven modulation of the positional eccentricity of the eyes through a VOR frequency-dependent gating mechanism[8] along with the residual effect of the vestibular modulation on the swim-related ocular motion magnitude through vestibulo-spinal pathways.

## Discussion

Locomotion-driven gaze-stabilizing eye movements appear concomitantly with visual- and otolith organ-driven ocular motor reflexes during early larval life in *Xenopus* but are further improved thereafter by a second step, related to the acquisition of semicircular canal sensitivity. With the appearance of a robust angular VOR, rhythmic ocular movements driven by locomotor CPG efference copies experience a gradual alteration in the modulatory influence on the impact of semicircular canal signals during larval development. During strong swimming with high-

**Fig. 6 Visuo-vestibulo-ocular motor computations. a** Schematic of the network model of brainstem and spinal circuit elements putatively involved in vestibulo- and spino-ocular motor signal integration during swimming. CPG, swim central pattern generator; dIN, descendant excitatory interneuron; cIN, commissural inhibitory interneuron; sp MN, spinal motoneuron; EC, efference copy neuron; VO, vestibulo-ocular; VS, vestibulo-spinal; Abd MN, abducens motoneuron. **b** Bilateral current injection of 0.15 nA into dIN started a 7 Hz fictive swim rhythm (light blue traces, sp MN; integrated traces in blue) and coupled ocular motor activity (light orange traces, Abd MN; integrated traces in orange). **c** Sinusoidal current injection at 1 Hz into VO neurons produced an aVOR in extraocular motoneurons (oranges traces, Abd MN). **d** The model robustly reproduced the observed variations of the eye motion magnitude and eye position eccentricity during the induced sinusoidal head rotation and tail undulation, respectively, simulated by sinusoidal currents injected into vestibular neurons (VO and VS) and current injection into dIN; large currents (> 0.16 nA), injected into dIN ($d_1$) produced swimming at high frequency and gating of VO neuronal signaling through an inhibitory current; smaller currents (< 0.16 nA) injected into dIN produced swimming at low frequency without inhibition of VO neuronal activity ($d_2$). **e** Scatter plot of fictive swim frequency, simulated as intracellular discharge of sp MN as function of the intensity of dIN current injection (dIN inj. cur.); the vertical dotted line represents the current injection magnitude (0.16 nA) above which VO neuronal activity becomes inhibited. **f** Scatter plots of Abd MN burst amplitude ($f_1$) and discharge baseline ($f_2$) variations (calculated from the integrated trace; see inset scheme in (**b**)) as function of the sp MN swimming frequency (swim freq). Two-tailed $p$ values were calculated to estimate the Pearson coefficient correlation (R) significance between Abd MN burst amplitude and sp MN swimming frequency, and between discharge baseline and sp MN swimming frequency (respectively, $p = 0.3324$ and $p = 0.0004$). Statistical significance is indicated as *** $p < 0.001$, $ns$ non-significant. Source data are provided as a Source Data file.

frequency tail undulations, preferentially occurring in young larvae, locomotor efference copies suppress the vestibulo-ocular motor consequences of horizontal head rotations. This limits the vestibular impact on eye movements to the modulation of swim-related ocular rotations around a rather invariant eccentric position of the eyes. Older larvae, which predominantly swim with low-frequency tail undulations, exhibit ocular motor reactions that result from a more inclusive interaction between horizontal head rotation signals and locomotor efference copies. The corresponding computational regime thereby produces gaze-stabilizing conjugate eye movements that are likely a combination between the angular VOR and spino-ocular motor coupling.

**Common developmental time course of sensory-based and locomotion-driven gaze-stabilization.** *Xenopus* larvae acquire the capacity to swim freely around stage 37/38 when spinal motor networks are sufficiently mature to allow the generation of motor commands for axial muscle contractions that produce rostro-caudally progressing undulations of the tail (Fig. 7[22,32–34]). However, first locomotion-driven eye movements were only observed at stage 42–43, despite the earlier onset of spinal CPG activity (Fig. 7). Interestingly the onset of locomotion-driven ocular motor behavior concurs with the onset of visual motion-driven ocular motor reflexes and otolith organ-dependent VOR components (Fig. 7[20,21]). While the vestibular sensory periphery continues to grow after hatching until late pro-metamorphic stages, the morphogenesis of the labyrinthine endorgans occurs early during ontogeny, concomitantly with central vestibulo-motor connections[35,36]. Based on data from chicken embryos, the first synapse to form along the three-neuronal VOR arc appears to be the connection between extraocular motoneurons and eye muscles[37]. Subsequently, vestibular afferent fibers connect to central vestibular neurons before the latter neurons form synaptic connections with extraocular motoneurons. This sequence complies with the onset of otolith organ-dependent reflexive eye movements. Hence, from stage 42 onward, the maturation of otolith organs allows detection of gravitoinertial head movements[16,21]. In contrast, vestibular projections to the spinal cord are fully established at stage 42–43[38]. Thus, the occurrence of first locomotion-driven eye movements could depend on established central otolithic VOR connections as well as on the functionality of the respective endorgans. However, if this logic reflects the full picture is unknown so far. In fact the developmental switch from a single spike spinal motor output to a burst pattern between stage 37/38 and stage 42 could be the origin of or at least supplement the increased spino-ocular motor coupling during this period (Supplementary Fig. 1a[34]).

**Locomotor and spino-ocular motor function depend on a joint developmental progression of otolith and semicircular canal organs.** After the onset of the aVOR, spinal CPG-driven eye movements become more efficient in the compensation of tail undulatory movements at all frequencies providing appropriate amplitude and temporal relationships (constant gain and minimal phase shift; see Fig. 2b, Supplementary Fig. 2). The inappropriate amplitude and the phase shift of both, locomotor strengths and locomotion-induced eye movements, in hyaluronidase-treated and thus semicircular canal-deficient larvae suggests that this second maturation step depends on the integrity of the ducts (Fig. 3). Unlike semicircular canal-related vestibulo-ocular connections where the different duct and eye muscle pairs share a coplanar spatial organization[28,39], macular VOR circuits have to map an otolithic planar sensitivity with a 360° directional distribution onto spatially appropriate sets of extraocular motoneurons and associated eye muscles with directionally matching pulling directions[40]. This major difference could explain the two steps in the maturation process of locomotion-induced ocular motor behaviors, before and after the emergence of the aVOR at stage 49/50. Therefore, given the absence of matching sensory and extraocular motor frameworks for otolithic signals, semicircular canal signals could be the relevant sensory elements to spatio-dynamically tune efference copy-based ocular motor behaviors at later developmental stages (Fig. 7). In fact, the spatial tuning of otolithic signals, transmitted by extraocular motoneurons, depends on a co-activation of specific epithelial sectors of the utricle, spatially aligned with a particular semicircular canal[26]. As supportive evidence for this assumption, the spatial tuning of utricular signals coincides with the appearance of the aVOR but is abolished in semicircular canal-deficient *Xenopus* larvae[26,41]. This spatial refinement of otolithic signals by semicircular canal signals could also assist in the observed semicircular canal-dependent second maturation step of locomotion-induced eye movements. However, at present, we can only speculate on the causality of this finding. Larger tail beat amplitudes evoke larger head deviations that require equally large eye movements. In aVOR-disabled animals, horizontal semicircular canals cannot assist in this task. Therefore, these animals would produce larger uncompensated residual error signals compared to controls. As a potential reactive mechanism, this might cause a reduction in tail oscillation amplitudes to minimize uncompensated eye movement components. In this case, the smaller eye movements would not result from a direct effect of absent semicircular canals but from an indirect consequence through smaller tail undulations. Such an impact of vestibular signals on the axial musculature has recently been demonstrated in larval zebrafish[42]. Maturation of

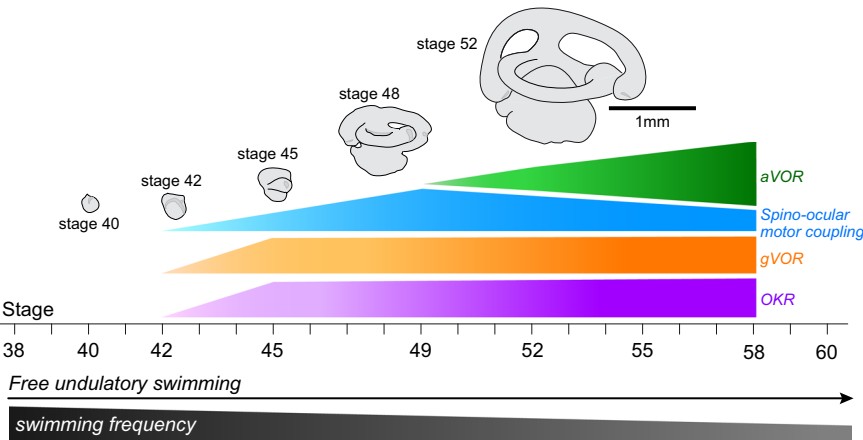

**Fig. 7 Developmental onset and progression of gaze-stabilizing eye movements.** Schematic depicting the ontogenetic onset and maturation of angular and gravitoinertial vestibulo-ocular reflexes (aVOR, gVOR), the optokinetic reflex (OKR) and swim-related spino-ocular motor coupling with respect to larval stages and ongoing growth of inner ear endorgans. Note the color-coding of the different sensory-motor components.

coordinated swimming in these animals requires vestibular sensory elements to be perfectly developed, demonstrated by the fact that vestibular-deficient mutants were unable to produce fin-body synergies during swimming[42].

**Vestibular computations responsible for different modes of locomotion-driven ocular motor behaviors.** The current data demonstrate that semicircular canal signals interact with locomotor efference copy-driven spino-ocular motor commands through different computational modes. Although both computational modes can be observed in a given animal, depending on the swim frequency, the differential integration is largely related to the developmental age based on the fact that the swim-related tail undulation frequency is a highly decisive parameter of larval locomotor ontogeny[27].

At stage 49–51 when the aVOR is not yet fully established and the animals express high-frequency swimming movements, horizontal head rotations have a limited impact on locomotor efference copy-induced eye movements. The role is restricted to the modulation of the swim-related ocular motion magnitude, to produce compensatory eye movements on a pure swim cycle-based phase coupling. This effect could result from a vestibular modulation of the CPG activity through vestibulo-spinal pathways, thus indirectly affecting CPG efference copy signaling without changing the temporal setting of a powerful spino-ocular motor coupling due to the high frequency swimming. These findings concur with previous observations demonstrating that the *lateral rectus* nerve discharge is gated by CPG efference copies during combined fictive swimming and horizontal head rotation at stage 52–53[8]. Therefore, integration of semicircular canal sensory inputs and locomotor commands, mediated by spino-ocular motor pathways during high frequency undulatory swimming, mainly present in young larvae, could result from a gating of VOR signals by efference copy signals with a residual modulation through the vestibulo-spinal control of spinal locomotor networks, as supported by our computational model (Fig. 6, Supplementary Fig. 5).

At older developmental stages when the VOR is fully mature and tadpoles express low-frequency tail undulations, horizontal head rotations have a more pronounced impact on locomotor efference copy-induced eye movements. Apart from a modulation of the swim-related ocular motion magnitude, angular vestibular motion signals also modulate the swim-related ocular position eccentricity with a temporal signature that reflects the timing of

the VOR. This latter feedback-feedforward interaction of motion signals during swimming effectively produces compensatory eye movements that derive from the combination of aVOR and locomotor activity-related ocular motor behaviors. Therefore, in older larvae, which generally exhibit low-frequency swimming movements, the sensory-motor integration results from a summation of aVOR signals and locomotor efference copy-driven spino-ocular motor commands, presumably fusing directly at the level of extraocular motoneurons as demonstrated earlier in empiric experiments[8] and which found entry into the computational model (see Fig. 6). If the locomotor activity-related attenuation of vestibulo-motor signals, observed during high-frequency swimming, is produced by an inhibitory action, exerted on horizontal aVOR commands at the level of central vestibular neurons or more directly at the level of extraocular motoneurons remains to be investigated. This swim frequency-dependent fusion of spino-ocular and vestibulo-ocular motor signals could be the result of a synaptic integration of signals transferred by the VOR circuit with those mediated by the ascending spino-ocular motor pathway, both of which are known to terminate directly on abducens motoneurons. Differentiating between the two mechanisms requires future experimental perturbations such as surgical or pharmacological manipulations, guided by predictions made by the computational model.

The principles of locomotor dynamics-related interactions between vestibular- and spinal CPG-driven eye movements in *Xenopus* might also govern optimal gaze-stabilization in terrestrial vertebrates including humans[14]. In fact, the predominance of spino-ocular motor behavior during rhythmic walking in humans appears to be a key feature for effective gaze-stabilization. However, in contrast to quadrupedal locomotion, the study of a simpler and more accessible model, such as larval *Xenopus*, allows demonstrating the ontogenetic establishment, maturation and tuning of associated feedforward gaze-stabilizing motor function. In a comparable manner, the implication of the vestibular sense has recently been demonstrated for the development of locomotor coordination and postural control[42,43]. While vestibular organs certainly play the dominant role as principal motion sensors in terrestrial vertebrates, aquatic anamniotes such as fish and amphibians might also recruit lateral line signals to extract information about motion in space[44]. So far, eye movements triggered by lateral line stimulation have not been demonstrated. However, the common synaptic input of ascending locomotor efference copy pathways to both, vestibular and lateral line efferent neurons[45,46], suggests that the two sensory systems might

also share common motor targets such as extraocular motoneurons.

## Methods

**Animals**. Experiments were conducted on larvae of the South African clawed toad *Xenopus laevis* of either sex ($N = 164$), obtained from the Xenopus Biology Resources Center (University of Rennes 1, France; http://xenopus.univ-rennes1.fr/). Animals were maintained at 20–22 °C in filtered aquarium water with a 12:12 h light/dark cycle. Experiments were performed on tadpoles between developmental stages 40 and 58, classified by stage-specific morphological features[47]. All procedures were carried out in accordance with and were approved by the University of Bordeaux ethics committee (#2016011518042273 APAFIS #3612).

**Video recordings and analysis of eye and tail movements**. The performance of ocular motor behaviors during passive head movements, visual image motion and undulatory swimming was recorded and quantified in semi-intact preparations[9] on an experimental setup from the Animotion collaborative core facility at the INCIA laboratory (CNRS UMR5287, Université de Bordeaux, http://www.incia.u-bordeaux1.fr/spip.php?rubrique193). To generate such preparations, tadpoles were anesthetized in an aqueous solution of 0.05% 3-aminobenzoic acid ethyl ester methane sulfonate (MS-222) and subsequently transferred to a dish containing oxygenated (95% $O_2$, 5% $CO_2$) Ringer solution (NaCl, 120 mM, KCl, 2.5 mM, $CaCl_2$ 5 mM; $MgCl_2$ 1 mM, $NaHCO_3$ 15 mM; pH 7.4) at ~18 °C. Isolation of the preparation was achieved by removal of the lower jaws and evisceration or decapitation at various, experiment-dependent levels of the spinal cord. The central nervous system, inner ears, eyes and motor effectors including efferent and afferent neuronal innervation, respectively, remained functionally preserved[48]. Following removal of the skin above the head, the cartilaginous skull was opened from dorsal to disconnect the forebrain and to expose the hindbrain thereby providing access for the Ringer solution during the submerged placement in the recording chamber. The preparation was firmly secured with the dorsal side up to a Sylgard-lined floor and was continuously superfused with oxygenated Ringer solution at a rate of 1.5–2.0 ml/min.

**Recordings of angular and gravitoinertial vestibulo-ocular reflexes**. Horizontal and vertical eye movements were elicited during application of head motion stimuli. For this purpose, the recording chamber was mounted onto a computer-controlled, motorized turn-table (Technoshop COH@BIT, IUT de Bordeaux, Université de Bordeaux). The preparation in the chamber was positioned such that the vertical rotation axis was centered in the midline between the bilateral otic capsules and the horizontal axis was located at the same dorso-ventral level as the otic capsules to ensure optimal activation of all semicircular canals and both utricles. Imposed horizontal head motion consisted of sinusoidal rotations around the vertical axis at a frequency of 1 Hz and maximal stimulus position eccentricity that ranged from ± 5° to ± 25°. Head roll motion stimulation consisted of sinusoidal rotations around a rostro-caudally directed horizontal axis at a frequency of 0.25 Hz and maximal stimulus position eccentricity that ranged from ± 2.5° to ± 20°. To exclude contributions of visuo-motor reflexes and locomotor efference copy-driven eye movements to the responses in particular subsets of the experiments, both optic nerves were transected at the level of the optic chiasm and the tail was severed at the level of the first spinal segments to eliminate an impact of ascending spino-ocular motor commands[8].

**Recording of optokinetic reflexes**. Horizontal visual motion-induced eye movements were evoked by a rotating large-field visual scene. Stimulation was provided by placing the recording chamber with the preparation in the center of a custom-made servo-controlled motorized drum with a diameter of 8 cm[49]. The wall of the rotating drum was illuminated from below, by visible light LEDs and displayed a pattern of equally spaced, vertical, black and white stripes with a spatial size of 15°/15°. The pattern was sinusoidally oscillated at a frequency of 0.25 Hz and maximal positional eccentricity of ± 2.5°–20°.

**Recording of eye movements during undulatory tail motion**. Horizontal eye movements were recorded during spontaneous episodes of left-right tail undulations[8]. Semi-intact preparations with intact tails were firmly secured by tight mechanical fixation of the head to the Sylgard floor of the recording chamber while the tail remained mobile to execute undulatory swimming movements. The solid mechanical fixation of the head prevented any residual head movements and thus the activation of vestibular reflexes, while potential optokinetic reflexes were excluded by the transection of both optic nerves (see above). Movements of the eyes and the tail during swimming and vestibular motion stimulation were video-recorded at 500 frames per second (fps) with a high-speed digital camera (Basler AG, ac1920) equipped with a micro-inspection lens system (Optem MVZL macro video zoom lens, QIOPTIQ). For the acquisition of horizontal eye movements, the camera was placed above the preparation; vertical eye movements during head roll motion were acquired with the camera positioned sideward of one eye. Tracking of the eyes and tail was automated using a custom-made software, programmed in Python 3.5 environment[9]. The software performed a frame-by-frame calculation of

the angles between the major axis of the oval-shaped eye and the longitudinal axis of the head, or the angle created by the positional deviation of the first five tail myotomes relative to the longitudinal axis of the head[9].

**Recording of extraocular and spinal motor nerve spike discharge**. Spike activity of ocular motor nerves and spinal ventral roots (Vr) were simultaneously recorded as described previously[8,24]. Respective preparations were produced as indicated above with the exception that the tail was not transversally severed but carefully dissected to expose the spinal cord along with the ventral roots until segment 22–25, while the adjoining distal part of the tail was anatomically preserved. Selected Vrs along the dissected spinal cord were identified for the recordings. The *lateral rectus* (LR) eye muscle-innervating abducens nerve was disconnected from its target muscle close to the neuro-muscular innervation site, cleaned from connective tissue and prepared for the recording with suction electrodes[8].

Extracellular spike discharge of the LR motor nerve and Vrs was recorded with glass suction electrodes, connected to a differential AC amplifier (A-M System Model 1700 AC, Carlsborg, US). Following production of the electrodes with a horizontal puller (P-87 Brown/Flaming, Sutter Instruments Company, Novato, CA, USA), tips were broken and individually adjusted to fit the LR nerve or Vr diameters. Electrophysiological signals were digitized at 10 kHz (CED 1401, Cambridge Electronic Design, UK), displayed and stored on a computer for offline analysis. In some experiments, recordings of the LR nerve of one eye were coupled with video recordings of the movements of the intact eye on the other side. In this experimental setting, eye movements were captured with a Basler camera (specification see above), placed above the eye and were tracked using custom-made software as described recently[9].

**Injection of hyaluronidase into the bilateral otic capsules**. The formation of semicircular canals during development was prevented according to a previously established method[25,26]. A single injection of a calibrated volume (10 nl) of the enzyme hyaluronidase (0.5 mg/mL, Sigma-Aldrich, France), dissolved in artificial endolymph solution was made into both otic capsules in larvae at developmental stage 44, prior to the formation of the semicircular canals. Thereafter, animals were allowed to further develop until usage in the respective experiments. As demonstrated, such tadpoles anatomically lack all semicircular canals as well as corresponding ocular motor reflexes, while the utricle and associated sensory-motor computations remain functionally preserved[16,26].

**Signal processing and data analysis**. Raw data of video sequences or electrophysiological recordings were processed off-line and analyzed with Dataview v11.5.1 (by W.J. Heitler, University of St Andrews, Scotland). For video image processing, traces of movements obtained from the eyes and/or tail were first filtered with a 20 Hz low-pass filter. Maximal excursions of the eyes were identified for each individual motion cycle and were used to calculate eye movement amplitudes. Tail-based swim cycles were defined by the repetitive angular excursion of the tail, with deviations from the null angle (longitudinal head/body axis) as indicator for the onset of each cycle. Maximal angular excursions of the tail and eyes (peak to peak amplitudes) were determined for each individual swim cycle and used to calculate the instantaneous peak frequency and temporal relationship between tail and eye movements. Head motion-evoked eye movement components during concurrent vestibular stimulation and fictive swimming were obtained by interpolation between each swim cycle-related average eye excursion. This procedure revealed locomotion-triggered eye motion magnitudes and dynamics, which would be otherwise masked by vestibulo-ocular response components. Variations of the eye motion magnitude/eccentricity during head motion were calculated for a minimum of four vestibular stimulus cycles. Traces with obvious stimulus-unrelated slow positional deviations were excluded from further analysis.

Spike discharge recordings of LR motor nerves and spinal Vrs were rectified and filtered (moving average, time constant: 25 ms, one iteration) to obtain the integral of the spike activity. Temporal parameters of fictive swim episodes were obtained from the timing of the spinal Vr burst discharge. An event was marked for each peak detected in the Vr and LR recordings indicating the phase relation between spinal and ocular motor spike activity during each swim cycle. Frequency and temporal relationships were subsequently calculated using the Dataview software.

**Spectral analysis**. The principal frequency content of eye movements and LR motor nerve activity, corresponding to fictive swimming *versus* vestibular stimulation, was extracted by spectral analyses. The task was performed with R software (v3.5.2, R Core Team, 2014; http://www.R-project.org/) and a custom-built R script from "waveletComp" package (Computational Wavelet analysis; R package v1.0; http://CRAN.R-project.org/package=WavelettComp)[50]. The spectral analysis consisted of two successive procedures, with a Fourier transformation (FFT) as first step, followed by a wavelet analysis. The FFT decomposed the recorded signal in multiple sinusoidal signals with different frequencies thereby identifying the principal oscillation frequencies. Thereafter, the wavelet analysis confirmed the principal oscillation frequencies and reconstructed sinusoidal signals corresponding to these frequencies. Oscillation frequencies usually ranged from 0.5 to 16 Hz. The results were presented as spectrograms providing the wavelet power spectrum of each continuous oscillation within the frequency range of the recordings.

Periodograms illustrated in addition the average power of each oscillation frequency. This required down-sampling of the original spike discharge recording to 500 Hz (performed with the Dataview software, interpolation method) in order to increase the computing velocity of the analysis algorithm.

**Computational model**. A computational neural network model was created with the NEURON 7.3 program[51]. Swim-related central pattern generator (swim CPG) circuitry with alternating rhythmicity was inspired by a previously established functional model[29,52]. The anatomical model, employed in the present study included 14 individual bilateral neurons constructed from 3 types of distinct anatomical template neurons (see code availability for more details) including a sensory neuron model (template "vest_spin" in the code, including vestibulo-ocular and vestibulo-spinal neurons), a generator neuron model (template "dIN" in the code) and a motor neuron model (template "MN" in the code, including spinal motoneurons – sp MN, commissural – cIN and efference copy - EC neurons). In contrast to the original approach, the present model simulated only one spinal segment. The simplified swim CPG network thus consists of two, bilateral symmetrical, swim circuits. Each contains an excitatory interneuron (dIN) forming excitatory synapses on cIN and sp MN of their respective side. Right and left swim circuits were coupled across the midline via reciprocal inhibitory connections through cIN. The resulting activity generates a bursting discharge in bilateral spinal motoneurons (sp MN), alternatingly activated by the CPG rhythm. This motoneuronal burst discharge simulates the known fictive swim pattern. The CPG rhythm is initiated by a current pulse (iStim) injected into both dINs with a 50 ms delay between the left and right side, triggering their successive and alternating depolarization. This dIN depolarization causes a subsequent activation of the ipsilateral cIN and an inhibition of contralateral cINs, dINs and sp MNs, respectively. The depolarization of dIN also activates its own AMPA and NMDA recurrent synapses, resulting in the depolarization of its membrane potential. This local network mechanism produces the self-rhythmicity of the CPG model. Accordingly, iStim currents of 0–0.3 nA generate swim frequencies of 0–12 Hz.

To model the spino-ocular motor coupling, the sp MN activates unilaterally an excitatory efference copy interneuron (EC) that projects to the contralateral abducens motoneuron (Abd MN). A vestibular neuron (VO) activates the contralateral Abd MN, representing the classical vestibulo-ocular projection pathway. A separate vestibular neuron (VS) activates both ipsilateral sp MN and EC neurons, thereby modeling a simple vestibulo-spinal pathway. Both VO and VS neurons can be individually or separately recruited by the injection of a continuous sinusoidal current with an adjustable intensity and frequency that mimics horizontal head rotation-related vestibular signals.

Activities simulated by sp MNs and Abd MNs are integrated (moving average of 15 ms, two iterations) with the Dataview software to compare the resultant motor output with the actual experimental data. The amplitudes of the integrated signals are therefore representative of the intra-burst spike frequency and based on the respective motoneuron membrane potential. To model the potential blocking of VO neuronal activity by swim-related activity, an inhibitory current (inhib. cur.) is also implemented in VO neurons. This latter inhibitory current is inspired by a previous model of inhibitory currents termed Inh, developed by Roberts and colleagues[29]. The inhibitory synaptic activity is set at a frequency of 10 Hz, with a conductance gGABA_Max indirectly scaled with the intensity of the iStim current via a sigmoidal function (synCoeff):

$$gGABA_{Max} = num_{syn_{Inh}} * Unitary\_gGABA * synCoeff$$

(where Unitary_gGABA = unitary synaptic conductance. Synaptic events are modeled using a two-exponential function with tau1 = 0.5 ms and tau2 = 100 ms) synCoeff is obtained from the sigmoidal steepness (sigmSteep) and amplitude (sigmAmp), that is adjustable. The setting of the sigmoidal mid-point value (iTrans) allows to adjust the level of swim activity (iCPG = mean value of the iStim current injected into both dINs) for which a transition reflex is observed:

$$sigmCoeff = sigmAmp * \left( \frac{\left(\exp\left(sigmSteep * (iCPG - iTrans)\right)\right)}{\exp\left(sigmSteep * (iCPG - iTrans)\right) + 1} \right)$$

With the implementation of this sigmoidal function, the strength of the inhibitory synapse increases rapidly and inhibits VO activity when the iCPG value raises above the iTrans value. The sigmoidal function as well as all others anatomical parameters are further specified in the code availability. The functional model including conductance properties is based on the Hodgkin-Huxley model and directly inspired by a previous published model[29,52] available in Model DB from https://senselab.med.yale.edu/ModelDB/ShowModel?model=238332&file=/tadpole-spinal-cord-models-master/functional%20model/#tabs-2. Details on our implemented functional model and parameter values are provided in the code availability. Parameter values can be set in the running Neuron software with a graphical user interface.

**Statistics**. After signal processing in Dataview, all data sets were analyzed with Prism7 (GraphPad, USA) and results were expressed as mean ± standard error of the mean (SEM), unless stated otherwise. Differences between two data sets were tested for significance using the unpaired two-tailed Mann-Whitney $U$-test, and the Kolmogorov-Smirnov test to compare between distributions. Multiple values were compared with the non-parametric Kruskal-Wallis test followed by a Dunn's multiple comparisons test. Correlations ($R$, Pearson correlation) and linear regression ($r^2$) was used to evaluate potential correlations between eye and tail movement amplitudes during fictive swimming, or between the ratio of eye/tail movements and tail excursion amplitudes with the latter as indicator for the strength of swimming. Differences between data sets were assumed to be significant at $p < 0.05$.

The temporal relationships between eye and tail movements at different developmental stages were assessed by a circular phase analysis of pooled data (Oriana 4.02 software (Kovach Computing Services, Wales)). The mean vector '$\mu$' and its length '$r$' indicated the preferred phase and strength of coupling, respectively. For non-uniform distributions (tested with the Rayleigh's uniformity test, $p$), mean phase values of each animal were plotted as the grand mean of individual means of phase relationships between eye and tail movements, expressed as ($\mu$, $r$, $p$). The preferred direction of the grand mean vector was tested by Moore's Modified Rayleigh test and the difference of phase relations between grand means at different developmental stages was evaluated by the Hotelling's two sample test.

**Reporting summary**. Further information on research design is available in the Nature Research Reporting Summary linked to this article.

## Data availability

Source data are provided with this paper and are available here: https://doi.org/10.17632/pgmb54b89b.2. All data presented in this article, including supplementary figures, are available and can be downloaded from this database. Further information and requests should be addressed to the corresponding author François M. Lambert (francois.lambert@u-bordeaux.fr). Source data are provided with this paper.

## Code availability

Computational modeling data are available here: https://doi.org/10.17632/m3dbv6m8bs.3. All part of the model presented in this article, including simulations shown in Fig. 6 and Supplementary Fig. 5, are available and can be downloaded from this database. Further information and requests should be addressed to the corresponding author François M. Lambert (francois.lambert@u-bordeaux.fr).

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

## Acknowledgements

F.M.L., G.C., and D.C. receive support from the French *Center National de la Recherche Scientifique*. This work was supported by grants from the *Agence National pour la Recherche* (Locogaze—ANR-15-CE32-0007-01, J.B.C.—D.C.—F.M.L.-M.B.), the Fondation pour la Recherche Médicale (DEQ20170336764, J.B.C.—D.C.—F.M.L.), the German Science Foundation (CRC 870 and STR 478/3-1, HS) and by funding from the LMU-Bordeaux Research Cooperation Program to F.M.L. and H.S.. FML received support from the INCIA CNRS UMR5287. MB received support from the Center National d'Etudes Spatiales and the Center National de la Recherche Scientifique. This study contributes to the IdEx Université de Paris ANR-18-IDEX-0001 (MB). The authors are grateful to Lionel Para-Iglesias for his work in the animal facility and for his help in 3D printing (*Creativ* core facility), to Philippe Chauvet from the INCIA mechanical workshop and to Daniel Cattaert for his help designing and improving the computational network model in the NEURON software. We thank Dr. Romuald Nargeot for his help with the statistical analyses of the data and the development of spectral analysis tools.

## Author contributions

Designed research: F.M.L.—D.C.—H.S., Performed research: F.M.L.—J.B.C.—G.C., Analyzed data: F.M.L.—J.B.C.—G.C.—M.B., Wrote the paper: F.M.L.—J.B.C.—D.C.—H.S.—M.B..

## Competing interests

The authors declare no competing interests.
