## [Peer review file · Nature Communications]

REVIEWER COMMENTS

Reviewer #1 (Remarks to the Author):

This study by Bacqué-Cazenave et al provides a detailed overview of the development of ocular corrections in *Xenopus* tadpoles and the various stimuli that induce them. The results describe the onset timing and effectiveness of different stimuli (visual, vestibular, CPG activity) for inducing corrective eye movements for gaze stability.

One of the main conclusions of the paper seems to be that the degree of vestibular-ocular corrections is diminished, to varying degrees, by the spinal efference copy-driven ocular pathway in a speed-dependent manner. While this in the opinion of this reviewer is an important finding, it is not clearly presented.

Major:

- 1) The organization of the data is sometimes confusing. For example, the different visual corrections described in Figure 1 do not seem to be directly related to the data presented later in the manuscript. Why is the data for aVOR, which is the main stimulus used later to explore interaction of spino-ocular coupling, not included in figure 1 with the other pathways for ocular movements? It could be useful to include these data together at the start of the manuscript so that they can be compared together.
- 2) Overall, it appears that the main factor underlying the degree of cancellation of vestibulo-ocular responses is the speed of locomotion (with an interaction with age). This is supported by the data showing different degrees of cancellation within an animal that depends on the speed of the locomotor CPG (as shown in Supp. Fig 3). This seems to be a major finding of the study and should be analyzed in more detail and included as a main figure. For example, in older animals (e.g. stage 53-58), is the degree of cancellation consistently correlated with the speed? Is it the absolute speed that determines the degree of cancellation or could it be the difference between the frequency of head motion vs. the CPG output frequency? This could be tested using different frequencies of head motion during swimming.
- 3) Throughout the manuscript the eye movements relating to vestibular motion are described as being cancelled, but they instead appear to be attenuated to varying degrees. However, there seems to be a persistent influence of horizontal head motion on eye movements across speeds and stages of development. This could be examined in more detail and clearly clarified in light of the previous studies showing a complete cancellation of vestibular eye movements by CPG efference copy driven eye movements – this appears to contradict the findings presented in the present study. Is the difference related to the age of the animals, the frequency of the head motion or locomotor output, or some other factor?
- 4) It is not clear if “cancellation” and “summation” is the correct way to interpret the data given that no direct mechanisms showing this is presented in this study.

5) The manuscript is, in many parts, written in a convoluted way and would gain in clarity if the authors would describe in a direct and simple way their findings and conclusions. For example, from reading the abstract it is difficult to understand what the paper is addressing.

Minor:

- 1) The correlation between tail amplitude and spino-ocular responses at different stages of development is shown. It would also be interesting to know how well aVOR ocular responses correlate with tail amplitudes at different stages. This has relevance for figure 3 where it seems that the reduced ocular responses in Hyalu tadpoles may simply be because of an effect of reduced tail amplitudes (hence reduced eye movements).
- 2) In figure 5 it would be informative to show an example of an LLR recording in a stage 48 animal (to match figure 4).
- 3) Is the principle of speed- and age-dependent cancellation of aVOR specific, or is there evidence of varying degrees of cancellation of other ocular pathways such as gVOR or OKR?
- 4) Elaborate on the responses seen in the VR recording in figure 5B. It seems that vestibular activation has an impact on the spinal CPG activity. Is this different from the results reported in the study published in Current Biology? And how this interaction affects the interpretation of the results.
- 5) Were the horizontal head movements always imposed at 1Hz? Could the degree of cancellation relate to the difference between head frequency and CPG frequency? This could be tested by varying the frequency of horizontal head motion.
- 6) In the discussion, supplementary figure 3 is referred to as supplementary figure 6.

Reviewer #2 (Remarks to the Author):

The study by Bacqué-Cazenave investigates concurrent development and interactions of gaze-stabilizing mechanisms. The manuscript is mostly well written and easy to follow. The authors propose a logical set of experiments to establish the timeline of different gaze-stabilizing motor behaviors and propose an interesting mechanism for the interaction of locomotor efference copy driven eye movements and later developing angular vestibulo-ocular reflexes. I enjoyed reading the manuscript but have the following two major concerns that should be addressed before publication.

Major concerns

1. The computational model is not sufficiently described for me to assess its validity. I can not find any information about the implementation of the model (ion channels, passive properties, synaptic connections, integration methods etc.).

You state in the method section that parameters were “specified in material availability.” But I can not find that information anywhere. I tried to follow the link given for data availability (<http://dx.doi.org/10.17632/pgmb54b89b.1>) but that only returns an error. I apologize if the code has been made available and I am just too stupid to find it, but I did try.

I also don't think that the model tests the main hypothesis that there is a frequency-dependent gaiting mechanism (Figure 7m,n and throughout the text).

In addition, Figure legend for Figure 8 states “the cancelation of vestibulo-ocular signals (b) and summation of vestibulo-ocular and spino-ocular signals (c) during swimming, depending on swim rhythm magnitudes.”

As I don't have the code, I can't confirm this. Is it really because of swim magnitude? What do you define as "magnitude" why didn't you implement a dependency on frequency as your data suggests?

As far as I can tell from the schematic and from your description, your model doesn't show a mechanisms for “magnitude” (or frequency) dependent gaiting, but the gaiting is just switched on or off dependent on your two binary simulation states. The conclusions you can draw from your model are therefore very limited.

Please add a full description of your model and make the code available. An exploration of frequency dependent mechanisms using the model would make the study even more powerful. Make sure that your conclusions from the model match the actual model.

2. On page 6 line 298 you state that: “the performance of these predictive eye movements was inferior to that of age matched controls and more comparable to that of younger larvae”

I don't think this statement is supported. You show in Fig3e that tail amplitude of aVOR-disabled animals was lower than controls and looking at the individual data in 3c it seems to me that eye movements follow the lower tail amplitudes with a similar correlation to that of controls that spontaneously show lower amplitudes.

If you wanted to show inferior performance/correlation of predictive eye movements probably some sort of correlation analysis would need to be done.

To me it sounds possible that predictive movements mature just fine, but eye movement amplitudes are lower because of lower tail amplitudes.

I am, however, intrigued about the fact that tail beat amplitude is reduced in the first place. Do you have any indication why tail beat amplitude stagnates and doesn't increase in aVOR-disabled animals? I would appreciate an addition to the discussion section to address this.

Minor concerns and typos:

Abstract, line 32-33: I don't think you established causality here, which "mutual" would imply (see my comments to the dataset presented in Fig 3.). I suggest removing mutual.

Page 4, line 201ff: the nomenclature for you model neurons is not consistent within your model and not consistent with common usage for different spinal CPG components.

- "cIN" is commonly used to denote commissural interneurons. Change this to "RG" (rhythm generator)
- "iEx": here you use i for interneuron, but used IN for interneuron in cIN, not consistent between the two. Also, are these really implemented as excitatory interneurons? Shouldn't these be reciprocally inhibitory commissural neurons to ensure left right alternation? My suggestion is to use either cIN (commissural interneuron) or i-cIN (inhibitory commissural interneuron)
- Change "Sp output" to "MN" or "sp. MN" if you want to distinguish between spinal and ocular MNs
- Make sure to make changes consistent throughout the text and in the figures

Page 6, line 291ff: this section would fit better under the next heading or maybe have it's own separate heading

Page 6, line 301: "had rather small amplitudes and high oscillation frequencies"

- is the increase in oscillation frequency significant when compared to control?

Page 6, line 307ff: I have no problem with your analysis and interpretation of the data in figure 5, but it is not an analysis of the influence of semicircular canals on locomotion-induced eye movements, but rather an analysis of the relative developmental changes.

For me this fit's better with the description of the developmental timeline that you have above.

I would consider moving this up and moving the experiments of Figure 3 down into this section.

In addition, if you aim is to suggest that the maturation of aVOR changes the motor efference copy driven eye movements, then an analysis of the locomotor power peak (like in 4e) should be shown as well.

page 7, line 334: "as well as on spinal central pattern generator performance."

you mention both effect on eye movement and spinal cpg performance, but then only speculate about a mechanism for ascending control of eye movements. It would be helpful for the reader if you could elaborate on the effects on spinal rhythm in the discussion.

Page 7, line 355-357: I assume optic nerves were cut here? Could you please clarify. Your methods section only states that they were cut in "particular subsets of the experiments". Also make this clear for all other experimental conditions.

Page 7, line 360ff: I found this section difficult to follow since you are jumping between figures. Would it be possible to arrange figure panels to fit the flow of your text? This is not a deal-breaker obviously, but might help other readers as well

Page 8, line 404: replace "bowls" with "spheres" (also in line 406, 408 and Fig legend Fig 7

Page 10, line 500: why is the model not part of the results section. I think it would be more appropriate to move it there.

Page 10, line 502: "During cancelation of vestibular signals (Fig. 8b) the direct vestibular input to abducens motoneurons was turned off"

was it just turned off, or did you implement a connection between EC and VO as you show in Fig 8b?

Fig legend 3, line 741: "ns" in legend but not indicated in figures

Fig legend 4: "Progressive alteration of semicircular canal influence on spino-ocular motor coupling during locomotion."

See major concern #2: This data doesn't show a causal relation, just a correlation, i.e. I don't think you have established that changes in aVOR influence spino-ocular motor coupling

Figure legend 8, line 820: "Abd MN" is only labeled as Abd in the figure

Figure legend 8, line 821: "robustly reproduced"

Did you test for robustness? If so, please report your tests and results.

Figure 8: what does gating mean here? Is this an inhibitory synapse onto VO? Or presynaptic inhibition? What does the open "gate" in the synaptic connection edge represent? I assume it's to indicate that the synapse becomes nonfunctional because of the inhibition from EC? Please clarify all this. There is an important difference between implementing an inhibitory synapse, or presynaptic inhibition, or just removing the synapse as you state somewhere else in the text.

Figure 8: why is the connection from EC to VO dashed?

Supplemental Figure 8: is it 6Hz for both conditions?

Supplemental Figure 8: Please distinguish between inhibitory connections and excitatory connections visually and within the legends (Same for Fig 8). Also, this figure doesn't show the "gating" mechanism that you have in Fig. 8.

Reviewer #3 (Remarks to the Author):

In this beautiful study the authors show the changing influences of locomotor activity and vestibular input in the maturation of gaze-stabilizing oculomotor movements in *Xenopus* tadpoles. Using a series of reduced preparations to deliver optokinetic and vestibular stimuli, tracking of eye movements, tail position, and electrophysiological recordings, they were able to dissect the various elements of the neural circuits underlying gaze stabilization during a key developmental time window in premetamorphic tadpoles.

During early stages spino-ocular coupling, optokinetic responses and otolith-mediated gVOR are the primary drivers of OM movements, while by later stages the angular VOR plays a more prominent role, as evidenced by altered OM in animals that had their semicircular canals ablated. Correlated to this is a developmental switch where OM compensation is initially primarily driven by efference copy signals from the spinal CPG, but then incorporated vestibular input to this signal. Nicely, the authors confirm their hypothesis with a computational network model.

Overall the experiments are of really high quality as is the data, showing a clear developmental time course and dependence on semicircular canal development. The various analyses are also well carried out and supported by the data. I think this is a remarkable example of how to breakdown various neural circuit elements within a neuroethological context as an animal goes through development. It should be of great interest to systems neuroscientists, computational neuroscientists, developmental neuroscientists, neuroethologists, and anyone interested in the development of neural circuits.

I only have a few minor comments, outlined below:

1. One key sensory influence on swimming behavior in *Xenopus*, in addition to the vestibular and visual inputs is the lateral line input, that senses water flow and vibration and likely to have a strong influence on oculomotor output during swimming. I realize these experiments were not done on freely swimming tadpoles, so examining the role of lateral line neuromasts on this process is not practical, however it is worth mentioning them as an additional possible influence in the paper's discussion.
2. I really think the computational model adds a nice element to the paper, and would suggest pulling it out of the supplementary figures and making it a more prominent part of the paper.
3. The paper is very dense and very hard to follow. I struggled a lot with it. Anything that can improve the flow would help make the paper more accessible to a wider range of readers.

-CA

Response to reviewers

All changes made in the revised version of the text are in blue.

REVIEWER COMMENTS

Reviewer #1 (Remarks to the Author):

This study by Bacqué-Cazenave et al provides a detailed overview of the development of ocular corrections in *Xenopus* tadpoles and the various stimuli that induce them. The results describe the onset timing and effectiveness of different stimuli (visual, vestibular, CPG activity) for inducing corrective eye movements for gaze stability.

One of the main conclusions of the paper seems to be that the degree of vestibular-ocular corrections is diminished, to varying degrees, by the spinal efference copy-driven ocular pathway in a speed-dependent manner. While this in the opinion of this reviewer is an important finding, it is not clearly presented.

REPLY: We have made a considerable effort to implement all comments of the reviewer into the new version of the manuscript. In particular:

1) we have reorganized the structure of the manuscript and added more data to make the starting point of our study clearer and to make our descriptions more accessible.

2) we have dropped the description “cancellation” versus “summation” to better describe the swim frequency-dependent differential interactions between locomotor efference copies and vestibular signals in the generation of compensatory eye movements.

We thus hope that the impact of swim strength (frequency/tail motion amplitude) on the eye eccentricity (formerly: position) and on the eye motion magnitude (formerly: amplitude) has become clearer.

Major:

1) The organization of the data is sometimes confusing. For example, the different visual corrections described in Figure 1 do not seem to be directly related to the data presented later in the manuscript. Why is the data for aVOR, which is the main stimulus used later to explore interaction of spino-ocular coupling, not included in figure 1 with the other pathways for ocular movements? It could be useful to include these data together at the start of the manuscript so that they can be compared together.

REPLY: Lambert et al., 2008 demonstrated that a functional aVOR in *Xenopus* tadpoles only appears at stage 49/50 due to the semicircular canal size limitation for motion detection. Prior to this stage, an aVOR could not be elicited. However, in order to incorporate the reviewer’s suggestion and to be consistent with the illustration of the eye movements at later developmental stages, we have now added sets of representative traces of aVOR eye movements of tadpoles at stages 40, 42 and 45 to Fig. 1, demonstrating the absence of aVOR responses in these young animals (see new Fig. 1d; see page 6 line 265-266).

2) Overall, it appears that the main factor underlying the degree of cancellation of vestibulo-ocular responses is the speed of locomotion (with an interaction with age). This is supported by the data showing different degrees of cancellation within an animal that depends on the speed of the locomotor CPG (as shown in Supp. Fig 3). This seems to be a major finding of the study and should be analyzed in more detail and included as a main figure.

GENERAL RESPONSE TO COMMENTS 2-4: Based on the comments of all three reviewers, we have

decided to replace the terms “cancellation” and “summation” throughout the entire manuscript. Instead, we have analyzed independently both the actual eye motion magnitude as well as the actual horizontal eccentricity of the eye around which the eye oscillates in the orbit (see representative schemes in new Fig. 5b₂-b₃); see page 8 lines 372-374.

Specific REPLY to 2): The reviewer is correct by assuming that locomotor strength is the main factor that determines the degree of modulation of the eye motion magnitude and eccentricity, both, within a given animal, because individuals are able to produce stronger or weaker (approximating faster and slower) swimming as well as along developmental stages, because younger animals express higher swim frequency-related tail oscillations than older animals (Hänzi and Straka, 2017). We have now analyzed this dependency in much greater detail as requested and accordingly included the data in a main figure (see new Fig. 5c₁-c₂ and new Supplemental fig. 4), which, in the first submission, was only presented in the third supplemental figure (originally as supplement to Fig. 6).

Given the fact that the head of the tadpole was mechanically secured to the floor of the recording chamber in our experiments, we are unfortunately unable to directly relate locomotor strength to the actual forward displacement of the animal and thus to the speed of swimming. Instead, the equivalent parameter of swim speed in our head-fixed preparations is the angular velocity of the tail motion. This angular velocity is linearly related to the forward propulsion (see Bacqué-Cazenave et al., 2018; Lambert et al., 2020). Thus, we determined the angular velocity of the tail excursions during swimming movements as a proxy for the forward velocity. This is possible because the angular velocity of the tail is linearly related to the swimming frequency (tail velocity: tail amplitude / period of the cycle, with the period = 1/ cycle frequency). Accordingly, eye movement parameters are presented in relation to the tail undulation amplitude, the only swimming parameter independent of the frequency.

Thus, we have taken the suggestion of the reviewer into account and produced a new analysis of vestibular-induced eye movement magnitudes and the modulation of the eye eccentricity relative to both tail amplitude and tail frequency, the two independent swim parameters in our experiments. The respective plots are now incorporated in the revised version as new Fig. 5c₁-c₂ and new Supplemental fig. 4a-c (see above). By using tail undulation frequency as proxy of swimming strength, we are now able to consistently relate our data, obtained in head-fixed preparations (mechanically secured to the floor), to real swimming behavior in freely locomoting larvae, as shown earlier (Bacqué-Cazenave et al., 2018; Lambert et al., 2020).

For example, in older animals (e.g. stage 53-58), is the degree of cancellation consistently correlated with the speed?

REPLY: In older animals the dependency of the attenuation of the VOR amplitude (newly termed: eccentricity) is consistently correlated with the swimming frequency but not with the tail undulation amplitude (see page 9 lines 385-389). This is now better illustrated in new Fig. 5 and new Supplemental fig. 4.

Is it the absolute speed that determines the degree of cancellation or could it be the difference between the frequency of head motion vs. the CPG output frequency? This could be tested using different frequencies of head motion during swimming.

REPLY: We have now, in addition to our previous data, also systematically analyzed the effect of the frequency of the imposed head rotation rather than the influence of the swimming frequency on locomotor-evoked eye movements by employing different head rotation stimuli between 0.5 Hz and 2 Hz as suggested by the reviewer. We now demonstrate that the influence of the vestibular stimulation on CPG efference copy-evoked eye motion magnitude is independent of the vestibular

stimulus frequency (0.5 Hz, 1 Hz, 2Hz; see new Fig. 5d₁-d₂ and new Supplemental fig. 4c₁-c₂).

3) Throughout the manuscript the eye movements relating to vestibular motion are described as being cancelled, but they instead appear to be attenuated to varying degrees. However, there seems to be a persistent influence of horizontal head motion on eye movements across speeds and stages of development. This could be examined in more detail and clearly clarified in light of the previous studies showing a complete cancellation of vestibular eye movements by CPG efference copy driven eye movements – this appears to contradict the findings presented in the present study. Is the difference related to the age of the animals, the frequency of the head motion or locomotor output, or some other factor?

REPLY: In Lambert et al., 2012, we described the cancellation of the horizontal VOR by spino-extraocular motor commands during fictive swimming. In this 2012 paper, we came to this conclusion based on the fact that during combined fictive swimming and horizontal head rotation (at 1 Hz), the lateral rectus discharge exhibited the same burst pattern as during fictive swimming alone, suggesting that any vestibular influence is suppressed by the locomotor signal. However, at that time, we did not analyze the individual burst magnitudes of the lateral rectus nerve discharge.

In the present study, we demonstrate that sinusoidal horizontal head rotations modulate only the ocular motion magnitude of the locomotor-induced eye movement during strong swimming. In contrast, the frequency of eye movements coincides with the swimming frequency. This impact on locomotor-induced eye motion magnitude corresponds to a modulation of the burst magnitude of the lateral rectus motor nerve discharge (see new Supplemental fig. 4e) without affecting the lateral rectus discharge frequency, which remains coupled to the swimming frequency.

This new result prompted us to re-analyze the electrophysiological recordings from our 2012 study. We found that the burst size of the lateral rectus nerve discharge was in fact also, though only slightly, modulated during fictive swimming with the frequency of the vestibular stimulus. Thus, the results in the present manuscript are not in contradiction with the past finding but, comply and further complement the conceptual organization of the signal processing. Accordingly, strong locomotor activity gates the aVOR. However, an effect of the aVOR on the locomotor-induced eye motion magnitude remains, potentially mediated by vestibulospinal pathways (see modeling of putative circuits in new Fig. 6).

4) It is not clear if “cancellation” and “summation” is the correct way to interpret the data given that no direct mechanisms showing this is presented in this study.

REPLY: We agree with the reviewer that “cancelation” and “summation” are terms, which inadequately describe the effects (see our GENERAL RESPONSE TO COMMENTS 2-4 above). Therefore, we replaced “cancelation” by “attenuation of the eye motion magnitude” and “summation” by “influence on the eye eccentricity”, terms that we feel better and more rigorously describe the effects of the vestibular stimulus on locomotion-induced eye movements. In addition, we have added a scheme in the new Fig. 5b that more clearly explains the two ocular motion parameters (magnitude and eccentricity) that are affected by the vestibular modulation to simplify and assist in the understanding of our results.

5) The manuscript is, in many parts, written in a convoluted way and would gain in clarity if the authors would describe in a direct and simple way their findings and conclusions. For example, from reading the abstract it is difficult to understand what the paper is addressing.

REPLY: We have restructured the manuscript to hopefully better transmit the novelty of our findings and to accommodate the criticisms and suggestions of all three reviewers. This also includes the abstract. We hope that our manuscript and the major results are now better accessible and easier to

follow.

Minor:

1) The correlation between tail amplitude and spino-ocular responses at different stages of development is shown. It would also be interesting to know how well aVOR ocular responses correlate with tail amplitudes at different stages.

REPLY: In order to isolate a pure aVOR response, independent of any other sources, aVOR-driven eye movements were measured in isolated preparations with both optic nerves transected (no visual inputs and thus no optokinetic responses) and the spinal cord fully sectioned at the first spinal segment to exclude potential tail proprioceptive signals as well as any residual effects of CPG efference copy signals. We have now incorporated a new plot (new Supplemental fig. 2f), where we illustrate the aVOR gain (ratio between stimulus motion and ocular motion) with respect to the tail oscillation magnitude (tail swim amplitude) across different developmental stages. This plot shows that the ocular motor performance is generally very similar between young (stage 49/50) and older animals (see page 6 lines 295-296).

This has relevance for figure 3 where it seems that the reduced ocular responses in Hyalu tadpoles may simply be because of an effect of reduced tail amplitudes (hence reduced eye movements).

REPLY: This addresses an important issue. It is currently not possible to clearly determine if the aVOR suppression in hyaluronidase-treated animals is directly due to an altered vestibular impact on locomotor-induced eye movements or derives from a reduced locomotor performance and consequently from a reduced efference copy signaling. Whatever the developmental impact of the aVOR suppression is, it is clear that the failure to form semicircular canals impairs the maturation of both swimming and locomotor-induced ocular motor behavior. We have now rephrased our statement on this aspect (see page 7, lines 314-322), according to the comments of reviewer.

2) In figure 5 it would be informative to show an example of an LLR recording in a stage 48 animal (to match figure 4).

REPLY: Recordings of the lateral rectus nerve with suction electrodes at developmental stage 48 are extremely difficult to perform because of the very small size of the nerve. We have succeeded to do so in a few animals during the collection of the data for the Lambert et al., 2008 paper where we demonstrated that the lateral rectus nerve discharge is not modulated during horizontal head rotation at stage 48. The absence of any horizontal head rotation-induced vestibular effects during swimming at stage 48, prior to the aVOR onset (Fig. 2 and Lambert et al., 2008), is now illustrated in Fig. 4a1-c1.

Due to this technical challenge, we therefore refrained from adding a figure illustrating the lack of vestibular influence in the absence of functional semicircular canals, despite the comprehensible suggestion of the reviewer. However, the new Fig. 1, as asked for by reviewer 1, now illustrates eye movements elicited by horizontal head rotation at stage 40, 42 and 45, demonstrating the absence of eye movements before aVOR onset at stage 49/50 as previously shown in Fig 2. In addition, the old figure 5, showing LR nerve discharge in response to combined vestibular stimulus and locomotion only served to illustrate the neural correlate of the ocular motor behavior shown in Fig. 4. According to the request of all reviewers and in order have the manuscript better accessible and easier to follow, we decided to transform old Fig. 5 into the new Supplemental fig. 3, given that the presented results are confirmatory additions to the main findings.

3) Is the principle of speed- and age-dependent cancellation of aVOR specific, or is there evidence of varying degrees of cancellation of other ocular pathways such as gVOR or OKR?

REPLY: Lambert et al., 2012 demonstrated that the gravito-inertial VOR (gVOR)-induced activity (activation of the utricle) and locomotor-induced activity was combined for the LR motor nerve discharge during combined head roll motion and fictive swimming. Moreover, extraocular motor nerves, involved in vertical and torsional eye movements, mainly activated by the gVOR (Lambert et al., 2008), were not attenuated by efference copy signaling. It thus seems that the gVOR and spinal motor commands use parallel pathways, with a final convergence at the level of extraocular motor nuclei where both signals are added. In addition, there is unpublished evidence from the Straka lab that during optokinetic stimulation, spino-ocular motor discharge and OKR-related discharge is also additively combined. However, no systematic investigation and respective data are yet available.

4) Elaborate on the responses seen in the VR recording in figure 5B. It seems that vestibular activation has an impact on the spinal CPG activity. Is this different from the results reported in the study published in Current Biology? And how this interaction affects the interpretation of the results.

REPLY: This is a good point. The authors agree that the vestibular activation has a limited impact on spinal CPG activity, as is reported in the first version of the manuscript (p.7 lines 333-336; p.7 lines 354-358; p.8 lines 414-416). Our hypothesis is that this vestibular modulation of the CPG activity could indirectly assist in the modulation of the locomotor-induced eye movement magnitude. This effect on CPG activity is probably mediated by vestibulospinal pathways, a hypothesis that the computational model in the manuscript was able to validate (see old Fig. 8b-c, now better illustrated in the new Fig. 6). However, the vestibular modulation of CPG activity is limited and rather variable, probably depending on swim strength.

5) Were the horizontal head movements always imposed at 1Hz? Could the degree of cancellation relate to the difference between head frequency and CPG frequency? This could be tested by varying the frequency of horizontal head motion.

REPLY: We have added the results from additional experiments, where different rotation frequencies were applied (0.5 Hz, 1 Hz, 2 Hz; see new Fig. 5 and Supplemental fig. 4). These new data demonstrate that the influence of the vestibular stimulus on locomotor efference copy-evoked eye movements is frequency independent (0.5 Hz, 1 Hz, 2 Hz); see page 8 lines 380-382.

6) In the discussion, supplementary figure 3 is referred to as supplementary figure 6.

REPLY: We have fixed this error.

Reviewer #2 (Remarks to the Author):

The study by Bacqué-Cazenave investigates concurrent development and interactions of gaze-stabilizing mechanisms. The manuscript is mostly well written and easy to follow. The authors propose a logical set of experiments to establish the timeline of different gaze-stabilizing motor behaviors and propose an interesting mechanism for the interaction of locomotor efference copy driven eye movements and later developing angular vestibulo-ocular reflexes. I enjoyed reading the manuscript but have the following two major concerns that should be addressed before publication.

Major concerns

1. The computational model is not sufficiently described for me to assess its validity. I can not find any information about the implementation of the model (ion channels, passive properties, synaptic connections, integration methods etc.).

You state in the method section that parameters were “specified in material availability.” But I can

not find that information anywhere. I tried to follow the link given for data availability (<http://dx.doi.org/10.17632/pgmb54b89b.1>) but that only returns an error. I apologize if the code has been made available and I am just too stupid to find it, but I did try.

REPLY: The authors apologize for this shortcoming. The final validation to have the material available online was not completed when the manuscript was submitted. This is why the computational model was not accessible to the Reviewer 2. After extending the model as suggested by the reviewer and making it part of the main manuscript (see below), it is now fully accessible online (<http://dx.doi.org/10.17632/m3dbv6m8bs.1>) to be download and tested by reviewer 2. Database doi are now reported in the DATA availability section of the manuscript (page 12 lines 616-617).

I also don't think that the model tests the main hypothesis that there is a frequency-dependent gaiting mechanism (Figure 7m,n and throughout the text).

REPLY: An extended and modified computational model has been provided (available here : <http://dx.doi.org/10.17632/m3dbv6m8bs.1>) for the revision that also tests the frequency dependency of the different integrating mechanisms. Results from this new model are shown in the new Fig. 6 and have been incorporated in the result section (see pages 9-10 lines 433-475) as also suggested by reviewer 3. The authors would like to point out that the original purpose of the computational model was not to test the effect of the swimming frequency but to rather test if the modulation magnitude (see responses to reviewer 1 and new Fig. 5b for magnitude definition) of the aVOR, which could be due to an indirect effect of vestibular activity, was potentially relayed through vestibulospinal pathways. This is why this model was originally placed in the discussion to illustrate the hypothesis of possible circuits involved in the feedback/feedforward vestibular/spinal interaction.

We have now made it more prominent as suggested by reviewer 2 and 3 by extending this part in the manuscript to allow testing the implication of vestibulospinal pathways and swimming frequency dependency by dedicating an entire paragraph to the respective modeling experiments in the result section; see pages 9-10 lines 433-475.

In addition, Figure legend for Figure 8 states "the cancelation of vestibulo-ocular signals (b) and summation of vestibulo-ocular and spino-ocular signals (c) during swimming, depending on swim rhythm magnitudes."

REPLY: The wording has been changed to accommodate the altered descriptions of the different integrative mechanisms.

As I don't have the code, I can't confirm this. Is it really because of swim magnitude? What do you define as "magnitude" why didn't you implement a dependency on frequency as your data suggests? As far as I can tell from the schematic and from your description, your model doesn't show a mechanism for "magnitude" (or frequency) dependent gaiting, but the gaiting is just switched on or off dependent on your two binary simulation states. The conclusions you can draw from your model are therefore very limited.

REPLY: We have now provided an extended model (new Fig. 6) that allows testing a potential frequency dependency (see above).

Please add a full description of your model and make the code available. An exploration of frequency dependent mechanisms using the model would make the study even more powerful. Make sure that your conclusions from the model match the actual model.

REPLY: This has been done in the online database (now accessible, see above) and the method

section where details of the improved model are now provided; see page 4-5 lines 203-224.

2. On page 6 line 298 you state that: “the performance of these predictive eye movements was inferior to that of age matched controls and more comparable to that of younger larvae”

I don't think this statement is supported. You show in Fig 3e that tail amplitude of aVOR-disabled animals was lower than controls and looking at the individual data in 3c it seems to me that eye movements follow the lower tail amplitudes with a similar correlation to that of controls that spontaneously show lower amplitudes.

REPLY: The reviewer is correct in pointing out this correlation. This aspect was initially discussed in the first version of the manuscript (Line 299-305). It is currently not possible to clarify if the hyaluronidase-induced reduced aVOR directly impacts locomotor-induced eye movements or reduces the locomotor performances (at the level of the spinal CPG) and thus indirectly impacts the spino-ocular coupling. Whatever the causality, it is, however, clear that the suppression of semicircular canal formation in hyaluronidase-treated animals impairs the maturation of both swimming and locomotor-induced ocular motor behavior. Accordingly, we have now altered the statement on this aspect (see page 7 lines 314-322).

If you wanted to show inferior performance/correlation of predictive eye movements probably some sort of correlation analysis would need to be done.

REPLY: As suggested by Reviewer 2, a correlation analysis on the data obtained from hyaluronidase-treated and normal stage 52 animals has been made. We found that the eye motion amplitude was correlated to the tail motion amplitude in both control and hyaluronidase stage 52. This analysis has been incorporated as a plot into the new Fig. 3.

To me it sounds possible that predictive movements mature just fine, but eye movement amplitudes are lower because of lower tail amplitudes.

REPLY: Based on our new analysis, this is well possible. We have now added the data to new Fig. 3 and added a corresponding statement on page 7 lines 314-318. See also response above.

I am, however, intrigued about the fact that tail beat amplitude is reduced in the first place. Do you have any indication why tail beat amplitude stagnates and doesn't increase in aVOR-disabled animals? I would appreciate an addition to the discussion section to address this.

REPLY: At the moment, we can only speculate on this aspect. Larger tail beat amplitudes evoke larger head deviations that require equally large eye movements. In aVOR-disabled animals, horizontal semicircular canals cannot assist in this task, thus producing larger uncompensated residual error signals as compared to controls. This potentially causes a reduction in tail oscillation amplitudes to minimize the uncompensated eye movement components. Additionally, it was recently demonstrated in larval zebrafish that the maturation of coordinated swimming requires vestibular sensory elements to be perfectly developed (Ehrlich and Schoppik, *Elife*, 2019). In this study vestibular deficient mutants were unable to produce fin-body synergies during swimming.

We have now added several sentences on this aspect to the discussion to accommodate this issue (see page 11 lines 532-542).

Minor concerns and typos:

Abstract, line 32-33: I don't think you established causality here, which “mutual” would imply (see my comments to the dataset presented in Fig 3.). I suggest removing mutual.

REPLY: We have removed the word “mutual”.

Page 4, line 201ff: the nomenclature for your model neurons is not consistent within your model and not consistent with common usage for different spinal CPG components.

- “cIN” is commonly used to denote commissural interneurons. Change this to “RG” (rhythm generator)
- “iEx”: here you use i for interneuron, but used IN for interneuron in cIN, not consistent between the two. Also, are these really implemented as excitatory interneurons? Shouldn't these be reciprocally inhibitory commissural neurons to ensure left right alternation? My suggestion is to use either cIN (commissural interneuron) or i-cIN (inhibitory commissural interneuron)
- Change “Sp output” to “MN” or “sp. MN” if you want to distinguish between spinal and ocular MNs
- Make sure to make changes consistent throughout the text and in the figures

REPLY: We have now made the nomenclature consistent within our manuscript, figures and model and also with abbreviations previously used in Roberts et al, 2014 to define the spinal CPG network in *Xenopus*. See scheme in new Fig. 6a and method section page 4-5 lines 203-224.

Page 6, line 291ff: this section would fit better under the next heading or maybe have it's own separate heading

REPLY: Given the extension of our modeling approach, we have now placed this part under the following headline in the results section: “*Impact of semicircular canals on the maturation of ocular motor behavior* “. (see page 6 line 304)

Page 6, line 301: “had rather small amplitudes and high oscillation frequencies” - is the increase in oscillation frequency significant when compared to control?

REPLY: Yes, both the swimming frequency and the tail amplitude are significantly different between stage 52 control and stage 52 hyaluronidase-treated (swimming frequency ctrl 52 vs hyalu 52 : * Mann-Whitney *U*-test, tail amplitude ctrl 52 vs hyalu 52 : ** Mann-Whitney *U*-test).

Page 6, line 307ff: I have no problem with your analysis and interpretation of the data in figure 5, but it is not an analysis of the influence of semicircular canals on locomotion-induced eye movements, but rather an analysis of the relative developmental changes. For me this fit's better with the description of the developmental timeline that you have above. I would consider moving this up and moving the experiments of Figure 3 down into this section.

REPLY: We assume that reviewer 2 refers to Fig. 4, because former Fig. 5 only illustrated the neural correlate of Fig. 4. If this is in fact the case, we agree with reviewer 2 that the data in Fig. 4 shows an analysis of the relative developmental changes. However, to maintain the flow of the data presentation, we decided to retain the previous sequence with the content of former Fig. 3 being illustrated prior to Fig. 4 for the following reason:

1. NEW Fig. 5 and 6 represent the direct continuation of Fig. 4 (we actually have now placed old Fig. 5 as NEW supplemental Fig. 3). In order to maintain the logic and to avoid restructuring the entire text by placing Fig. 3 after Fig. 4 and before Fig. 5 and 6, we decided to keep the previously established sequence. However, we streamlined the text and made our logic more accessible.

2. The analysis on which Fig. 3 is based on, was conducted in the same way that current Fig. 2 has been based on, to maintain the same interpretation logic between the two figures.

3. Accordingly, The logical flow of the paper is the following:

1. Fig. 2 shows the co-maturation of aVOR and efference copy signaling.
2. Fig. 3 demonstrates the locomotor behavior and the dependency of efference copy signaling on VOR maturation.
3. Fig. 4, 5, 6 illustrate the quantification of the interaction between VOR and efference copy signaling.

In addition, if your aim is to suggest that the maturation of aVOR changes the motor efference copy driven eye movements, then an analysis of the locomotor power peak (like in 4e) should be shown as well.

REPLY: Done as suggested. See analysis now in new Fig. 4f and page 7 lines 349-351.

page 7, line 334: “as well as on spinal central pattern generator performance.” you mention both effect on eye movement and spinal cpg performance, but then only speculate about a mechanism for ascending control of eye movements. It would be helpful for the reader if you could elaborate on the effects on spinal rhythm in the discussion.

REPLY: A discussion on the effects on spinal rhythm has now been added. See page 11 lines 550-561.

Page 7, line 355-357: I assume optic nerves were cut here? Could you please clarify. Your methods section only states that they were cut in “particular subsets of the experiments”. Also make this clear for all other experimental conditions.

REPLY: Yes, both optic nerves have been cut. This is now stated explicitly in the text (see page 3 lines 115-117 and lines 131-132).

Page 7, line 360ff: I found this section difficult to follow since you are jumping between figures. Would it be possible to arrange figure panels to fit the flow of your text? This is not a deal-breaker obviously, but might help other readers as well.

REPLY: We have rearranged the respective figure panels (see new Fig. 5 and supplemental Fig. 4) to increase the readability and accessibility of the text and figures (see pages 8-9 lines 369-432).

Page 8, line 404: replace “bowls” with “spheres” (also in line 406, 408 and Fig legend Fig 7)

REPLY: Done.

Page 10, line 500: why is the model not part of the results section. I think it would be more appropriate to move it there.

REPLY: The extended model is now part of the result section (see new Fig. 6).

Page 10, line 502: “During cancelation of vestibular signals (Fig. 8b) the direct vestibular input to abducens motoneurons was turned off”; was it just turned off, or did you implement a connection between EC and VO as you show in Fig 8b?

REPLY: This has been better specified now in the new model (new Fig. 6; see methods section pages 4-5 lines 203-224). So far, we are sure about one fact: the aVOR gating mechanism depends on CPG swimming activities (Lambert et al., 2012). However the neural mechanism (inhibitory neuron, pre-synaptic inhibition) underlying in this aVOR gating remains unknown. Given this statement, an

inhibitory synapse has been implemented in VO neurons and its intensity was fitted with intensities of all CPG neuron currents (iCPG). A sigmoidal function has been implemented into this inhibitory synapse with a steepness and an amplitude fixed by the user. When iCPG crossed the inflexion point of the sigmoidal function, VO neurons were consequently inhibited. The inflexion point represents the current injected into dIN to start the swimming rhythm in simulated CPG network. The equations and more details were added to the code, in the part “scaling inhibitory VO synapses”.

Fig legend 3, line 741: “ns” in legend but not indicated in figures

REPLY: This is now indicated also in the figures

Fig legend 4: “Progressive alteration of semicircular canal influence on spino-ocular motor coupling during locomotion.” See major concern #2: This data doesn't show a causal relation, just a correlation, i.e. I don't think you have established that changes in aVOR influence spino-ocular motor coupling

REPLY: This is correct. We have altered the wording accordingly; see page 7 line 314-322.

Figure legend 8, line 820: “Abd MN” is only labeled as Abd in the figure

REPLY: This part of Fig. 8 has been removed. Changes have been made to the new Fig. 6 to be consistent with the description in the text.

Figure legend 8, line 821: “robustly reproduced” Did you test for robustness? If so, please report your tests and results.

REPLY: We did not test for robustness but used the word “robustly” only to indicate a qualitatively consistent reproduction of the effect. We have now changed the wording.

Figure 8: what does gating mean here? Is this an inhibitory synapse onto VO? Or presynaptic inhibition? What does the open "gate" in the synaptic connection edge represent? I assume it's to indicate that the synapse becomes nonfunctional because of the inhibition from EC? Please clarify all this. There is an important difference between implementing an inhibitory synapse, or presynaptic inhibition, or just removing the synapse as you state somewhere else in the text.

REPLY: We have now clarified this aspect by adding several sentences at pages 9 lines 445-447 and page 11 lines 571-574. We do not refer to the imprecise ‘gating’ term anymore in the modeling section. The blocking of the VO neuron in the model is now implemented by a simple model of inhibition: an inhibitory current related to the CPG activity as described above (see response to minor comment page 10, line 502).

By the way, the detailed comprehension of the neural mechanism underlying the CPG-induced aVOR gating constitutes a full study by itself. We did not intend to explore this question more extensively in the present manuscript. This will be the topic of a future study with the further update of our computational model.

Figure 8: why is the connection from EC to VO dashed?

REPLY: This has been altered in the new, upgraded model (see new Fig. 6).

Supplemental Figure 8: is it 6Hz for both conditions?

REPLY: This aspect has been altered in the new, upgraded model (see new Fig. 6).

Supplemental Figure 8: Please distinguish between inhibitory connections and excitatory connections visually and within the legends (Same for Fig 8). Also, this figure doesn't show the "gating" mechanism that you have in Fig. 8.

REPLY: This aspect has been covered by alterations made in the new, upgraded model.

Reviewer #3 (Remarks to the Author):

In this beautiful study the authors show the changing influences of locomotor activity and vestibular input in the maturation of gaze-stabilizing oculomotor movements in *Xenopus* tadpoles. Using a series of reduced preparations to deliver optokinetic and vestibular stimuli, tracking of eye movements, tail position, and electrophysiological recordings, they were able to dissect the various elements of the neural circuits underlying gaze stabilization during a key developmental time window in premetamorphic tadpoles.

During early stages spino-ocular coupling, optokinetic responses and otolith-mediated gVOR are the primary drivers of OM movements, while by later stages the angular VOR plays a more prominent role, as evidenced by altered OM in animals that had their semicircular canals ablated. Correlated to this is a developmental switch where OM compensation is initially primarily driven by efference copy signals from the spinal CPG, but then incorporated vestibular input to this signal. Nicely, the authors confirm their hypothesis with a computational network model.

Overall the experiments are of really high quality as is the data, showing a clear developmental time course and dependence on semicircular canal development. The various analyses are also well carried out and supported by the data. I think this is a remarkable example of how to breakdown various neural circuit elements within a neuroethological context as an animal goes through development. It should be of great interest to systems neuroscientists, computational neuroscientists, developmental neuroscientists, neuroethologists, and anyone interested in the development of neural circuits.

REPLY: We thank the reviewer for the very positive assessment of our manuscript.

I only have a few minor comments, outlined below:

1. One key sensory influence on swimming behavior in *Xenopus*, in addition to the vestibular and visual inputs is the lateral line input, that senses water flow and vibration and likely to have a strong influence on oculomotor output during swimming. I realize these experiments were not done on freely swimming tadpoles, so examining the role of lateral line neuromasts on this process is not practical, however it is worth mentioning them as an additional possible influence in the paper's discussion.

REPLY: We have taken the reviewer's suggestion into account and indicate now the potential role of motion-induced lateral line signals as another sensory source for eliciting compensatory eye movements during locomotion. See now page 12 lines 587-592.

2. I really think the computational model adds a nice element to the paper, and would suggest pulling it out of the supplementary figures and making it a more prominent part of the paper.

REPLY: We have now extended the computational model and have made it a more prominent part of the manuscript as suggested by adding a full paragraph of descriptions to the result section (see page

9-10 lines 433-475) and illustrating the consequences of the modeling as a main figure (new figure 6), also in accordance with the comments of reviewer 2.

3. The paper is very dense and very hard to follow. I struggled a lot with it. Anything that can improve the flow would help make the paper more accessible to a wider range of readers.

REPLY: We have restructured the manuscript with the attempt to make it more accessible. We hope that we have now presented the major results in a clearer sequence and with a better wording. We also hope that by this rewriting we have increased the readability of the manuscript.

REVIEWER COMMENTS

Reviewer #1 (Remarks to the Author):

The authors have addressed all the comments and concerns, the revised manuscript is now acceptable for publication.

Reviewer #2 (Remarks to the Author):

I commend the authors for their extensive reworking of the manuscript. I think it is greatly improved.

My main concern with the revised version of the manuscript is missing information and a description of the model.

Modeling files seem available but extracting the downloaded zip file returns an error. After downloading all individual files, the code runs, but detailed instructions on how to recreate the figure panels are missing.

I looked through individual files in the repository, but to understand the model architecture, individual neuron properties, synaptic implementation, etc., the model including equations and parameters should be described in the manuscript. See also instructions here: <https://media.nature.com/full/nature-cms/documents/GuidelinesCodePublication.pdf>

It is for example not even clear to me if the neurons shown in the schematic are individual neurons or populations, how the rhythm is generated, and many other aspects of the model.

For example (just to illustrate that it was difficult for me to follow what you did in the model):

1) You write that “The activation of AMPA and NMDA synapses transforms the dINs into pacemaker neurons”, but you seem to inject a rhythmic current into dINs to drive them into rhythmicity (e.g. Fig 6b states “7 Hz current injection in dIN and the network seems to produce a 7Hz rhythm)). Then in Fig 6e you seem to inject continuous current at different strengths. If you DO indeed have pacemaker properties, how do these come about?

2) What does “currents generated by all CPG neurons (iCPG)” mean?

3) You implemented an inhibitory current that depends on stim intensity in a sigmoidal manner. Is that correct? Why then is there a sharp line indicating an on/off kind of mechanism in Figure 6e?

This is quite confusing, please clarify.

Typos

- Page 5 line 224: "material availability" doesn't exist as a section. The link to the code should be given under the heading "Code availability" (<https://www.nature.com/nature-portfolio/editorial-policies/reporting-standards#availability-of-computer-code>)

- Figure 6 b: the trace for Vest. sine is missing

- Figure 6 b-d1: label for the scale bar for sp MN int. is missing.

Reviewer #3 (Remarks to the Author):

The new version is much improved, both in terms of readability, but the addition of tada from younger tadpoles completes the picture better, as well as adding the model to the main paper. I think this is a great paper and I have no additional concerns.

REVIEWER COMMENTS

Reviewer #1 (Remarks to the Author):

The authors have addressed all the comments and concerns, the revised manuscript is now acceptable for publication.

REPLY: We thank the reviewer for the very positive final assessment of our manuscript.

Reviewer #2 (Remarks to the Author):

I commend the authors for their extensive reworking of the manuscript. I think it is greatly improved.

REPLY: We thank the reviewer for the very positive comment on the revision of our manuscript.

My main concern with the revised version of the manuscript is missing information and a description of the model.

REPLY: The computational model method section has been further extended to complete the description of the model including anatomical and functional aspects with the aim to answer the reviewer's comments listed below. The repository was also updated accordingly. Please find here the link to the site: <http://dx.doi.org/10.17632/m3dbv6m8bs.2>. We cleaned unused files and added a quick-start guide to run the code and to reproduce the panels of figure 6.

Modeling files seem available but extracting the downloaded zip file returns an error. After downloading all individual files, the code runs, but detailed instructions on how to recreate the figure panels are missing.

REPLY: The authors apologize for this hassle. We noticed that smooth running of the algorithms depends on the operating system. Using a windows system, it is necessary to generate as single DLL file from the NEURON software to run the code. Presumably, we suspect that the generic DLL in the previous repository was not functional on all computer systems.

We have now added more instructions to the repository to install NEURON and to generate the DLL file, adapted for different computer configurations. We also have added now a quick-start guide explaining how to use the NEURON graphical user interface and to load parameter template files corresponding to panels b-d in figure 6.

I looked through individual files in the repository, but to understand the model architecture, individual neuron properties, synaptic implementation, etc., the model including equations and parameters should be described in the manuscript. See also instructions here: <https://media.nature.com/full/nature-cms/documents/GuidelinesCodePublication.pdf>

It is for example not even clear to me if the neurons shown in the schematic are individual neurons or populations, how the rhythm is generated, and many other aspects of the model.

REPLY: To improve the understanding of the architectural code (anatomical and functional properties), the tree structure of the folder containing all files of the model is now described in the quick-start guide that has been added to the repository. In brief, our model is based on a previously

published model by Roberts et al., 2014 that was further complemented by Ferraro et al., 2017. Our anatomical model architecture differed from the original model published in these two papers because we only modeled a single central pattern generator (CPG) segment. Consequently, individual neurons are represented by our model. This is now clearly specified in the manuscript. However we used the same synaptic properties as previously described in the original model (see the following link): <https://senselab.med.yale.edu/ModelDB/ShowModel?model=238332&file=/tadpole-spinal-cord-models-master/functional%20model/#tabs-2>.

These neuronal synaptic properties are now also available in our code repository (functional properties). In order to keep the corresponding method section as simple and accessible as possible, we avoided to incorporate excessive numbers of known equations that have been already published. Rather, we focused on adding our implemented news features on the inhibition of vestibular neurons.

For example (just to illustrate that it was difficult for me to follow what you did in the model):
1) You write that “The activation of AMPA and NMDA synapses transforms the dINs into pacemaker neurons”, but you seem to inject a rhythmic current into dINs to drive them into rhythmicity (e.g. Fig 6b states “7 Hz current injection in dIN and the network seems to produce a 7Hz rhythm”). Then in Fig 6e you seem to inject continuous current at different strengths. If you DO indeed have pacemaker properties, how do these come about?

REPLY: We apologize for the misleading description and wording and the lack of sufficient and appropriate explication about the rhythm generating mechanism. The dIN neurons do not express pacemaker properties but the self-rhythmicity is caused by a reciprocal inhibition from cIN neurons and self-activated AMPA/NMDA synapses (see scheme in Fig. 6a). A more detailed description of this rhythmicity mechanism is now presented in the manuscript at lines 218-227. The current, used to trigger the swim rhythm derived from a pulse injected into dIN neurons with a 50 ms delay between the left and right side in order to initiate the self-rhythmicity.

2) What does “currents generated by all CPG neurons (iCPG)” mean?

REPLY: Again, the authors apologize for the misleading sentence. The iCPG current (defined now in the method section) is the average of the iStim current (also defined in the method section), which is injected simultaneously into both dIN neurons. This is now better specified in the manuscript at lines 222-223.

3) You implemented an inhibitory current that depends on stim intensity in a sigmoidal manner. Is that correct? Why then is there a sharp line indicating an on/off kind of mechanism in Figure 6e? This is quite confusing, please clarify.

REPLY: At present, the inhibitory mechanism that attenuates the activity in the vestibulo-ocular pathway during high swimming frequencies remains unknown. Therefore, in the absence of clear evidence, we implemented the simplest way to initiate the swim frequency-dependent inhibition of the VO neuron. In our model, the swim frequency is thus simply correlated with the intensity of the iStim current (Fig. 6e) and consequently with the intensity of the iCPG current (see question 2). In addition, the degree of inhibition of the VO neuron depends on the inhibitory synaptic activity implemented on the VO neuron. This synaptic activity is scaled with the iStim current intensity through a sigmoidal function. Using this simple mechanism, we are able to block the VO neuron

above a defined threshold (= i_{Trans} value, defined in the method section) of the i_{Stim} current, which eventually derives from the swim frequency. This equation is now stated in the manuscript and all indicated parameters are incorporated in the scaling of the inhibitory VO synapse section in the code file (XenopeCPG09d.hoc).

The inhibition of the VO neuron is not an ON/OFF mechanism but due to the fact that we have implemented a relatively sharp rise in the sigmoidal function this produces a rapid activation of this inhibition. To avoid misunderstanding, we now changed the graphical representation in Fig. 6e by drawing a graded “level of grey” bar attempting to mimic the respective degree of inhibition of the VO neuron. The rapid switch between the light area, (VO neuron not inhibited) and the dark area (VO neuron inhibited) occurs around 0.16 nA.

Typos

- Page 5 line 224: “material availability” doesn’t exist as a section. The link to the code should be given under the heading “Code availability” (<https://www.nature.com/nature-portfolio/editorial-policies/reporting-standards#availability-of-computer-code>)

REPLY: This has been changed and is now also in accordance with the Nature Communication policy.

- Figure 6 b: the trace for Vest. sine is missing

REPLY: The missing vestibular sinewave has been added to Fig. 6b.

- Figure 6 b-d1: label for the scale bar for sp MN int. is missing.

REPLY: The missing label for the scale bar of sp MN has been added to Fig. 6.

Reviewer #3 (Remarks to the Author):

The new version is much improved, both in terms of readability, but the addition of data from younger tadpoles completes the picture better, as well as adding the model to the main paper. I think this is a great paper and I have no additional concerns.

REPLY: We thank the reviewer for the very positive final assessment of our manuscript.

REVIEWERS' COMMENTS

Reviewer #2 (Remarks to the Author):

I thank the authors for improving their model documentation.

I am mostly satisfied with the changes to the manuscript, but ask the authors to fix the following before publication:

1: Your response states that there is no rhythmic injection into dINs and the rhythmicity is due to a network property, but in Figure 6b, d1, and d2 the labels say “7Hz current injection in dIN”, “8Hz cur. inj. in dIN”, and “6Hz cur. inj. in dIN”. Similar for Supplemental Figure 5. The figure legends also mention rhythmic current injection. This is wrong, right? Are you injecting a tonic current or not? If you are injecting rhythmic current, then is this what paces the rhythm and not the network properties? Please check also if there is similar wording somewhere else. Please correct, this is confusing.

2: When downloading the zip file, the subfolder Xenope09 is empty. I was able to download the individual files from the website, but you should fix this.

Your instructions make it sound as if there should be a nrvmodl.bat file in the nrvMod folder, but that wasn't the case for me. After compiling mod files from Neuron the code runs for me.